

# True eddy accumulation trace gas flux measurements: proof-of-concept

Lukas Siebicke[1] and Anas Emad[1]

[1]Bioclimatology, Georg-August-University of Goettingen, Goettingen, Germany

*Correspondence to:* Lukas Siebicke (Lukas.Siebicke@uni-goettingen.de)

**Abstract.** Micrometeorological methods to quantify fluxes of atmospheric constituents are key to understanding and managing the impact of land surface sources and sinks on air quality and atmospheric composition.

Important greenhouse gases are water vapor, carbon dioxide, methane, and nitrous oxide. Further important atmospheric constituents are aerosols which impact air quality and cloud formation, and volatile organic compounds. Many atmospheric
constituents therefore critically affect the health of ecosystems, and humans as well as climate.

The micrometeorological eddy covariance (EC) method has evolved as the method-of-choice for $CO_2$ and water vapor flux measurements using fast-response gas analyzers. While the EC method has also been used to measure other atmospheric constituents including methane, nitrous oxide, and ozone, the often relatively small fluxes of these constituents over ecosystems are much more challenging to measure by eddy covariance than $CO_2$ and water vapor fluxes. For many further atmospheric
constituents, eddy covariance is not an option due to the lack of sufficiently accurate and fast-response gas analyzers.

Therefore, alternative flux measurement methods are required for the observation of atmospheric constituent fluxes for which no fast-response gas analyzers exist or which require more accurate measurements. True eddy accumulation (TEA) is a direct flux measurement technique capable of using slow-response gas analyzers. Unlike its more frequently used derivative, known as the relaxed eddy accumulation (REA) method, TEA does not require the use of proxies and is therefore superior to the
indirect REA method.

The true eddy accumulation method is by design ideally suited for measuring a wide range of trace gases and other conserved constituents transported with the air. This is because TEA obtains whole air samples and is, in combination with constituent-specific fast or slow analyzers, a universal method for conserved scalars.

Despite the recognized value of the method, true eddy accumulation flux measurements remained very challenging to per-
form as they require fast and dynamic modulation of the air sampling mass flow rate proportional to the magnitude of the instantaneous vertical wind velocity. Appropriate techniques for dynamic mass flow control have long been unavailable, preventing the unlocking of the TEA method's potential for more than 40 years.

Recently, a new dynamic and accurate mass flow controller which can resolve turbulence at a frequency of 10 Hz and higher has been developed by the author. This study presents the proof-of-concept that practical true eddy accumulation trace gas flux
measurements are possible today using dynamic mass flow control, advanced real-time processing of wind measurements, and fully automatic gas handling.



We describe setup and methods of the TEA and EC reference flux measurements. The experiment was conducted over grassland and comprised seven days of continuous flux measurements at 30-min flux integration intervals. The results show that fluxes obtained by TEA compared favourably to EC reference flux measurements with coefficients of determination of up to 86% and a slope of 0.98.

We present a quantitative analysis of uncertainties of the mass flow control system, the gas analyzer and gas handling system and their impact on trace gas flux uncertainty, the impact of different approaches to coordinate rotation and uncertainties of vertical wind velocity measurements.

Challenges of TEA are highlighted and solutions presented. The current results are put into context of previous works. Finally, based on the current successful proof-of-concept, we suggest specific improvements towards long-term and reliable

true eddy accumulation flux measurements.

## 1   Introduction

The ability to observe the exchange of trace gases between the earth's surface and the atmosphere is key to understanding the functioning of ecosystems. Trace gas flux measurements allow quantifying how natural and anthropogenic systems affect atmospheric composition.

Many studies over the past decades have observed carbon dioxide ($CO_2$) and water vapor fluxes at ecosystem scale using micrometeorological methods (Baldocchi et al., 1988). Eddy covariance (EC) (Baldocchi, 2003, 2014) has become the most widely used method for measuring turbulent fluxes. Today the EC method is routinely being applied the world over including major flux networks FLUXNET, ICOS, and NEON.

The EC method requires fast-response gas analyzers which only exist for a few trace gas species, above all $CO_2$ and water

vapor but more recently also other trace gases, including methane ($CH_4$) and nitrous oxide ($N_2O$). However, for a large number of trace gases and atmospheric constituents, the applicability of the EC method is limited by lack of fast-response gas analyzers, by the high power demand necessary for sustaining high sample flow rates in some closed-path gas analyzer systems and by a possibly small signal-to-noise ratio of high frequency measurements.

A number of alternative turbulent flux measurement methods exist which can use slow-response gas analyzers, and might

provide more accurate results than eddy covariance with fast-response analyzers. These methods are applicable to a wide range of conserved trace gases, isotopes, aerosols, volatile organic compounds and other atmospheric constituents. An overview on selected micrometeorological methods applicable to slow-response gas analyzers follows, presenting the air sampling principles and timings, and stating advantages and disadvantages of each method.

True eddy accumulation (TEA) is an alternative to the EC method. Unique properties of the TEA method are highlighted

which make TEA stand out from other methods. This study is a contribution towards a practical implementation of the TEA method.



## 1.1 Micrometeorological methods suitable for slow-response gas analyzers

### 1.1.1 True eddy accumulation (TEA)

True eddy accumulation (Desjardins, 1977; Hicks and McMillen, 1984), refers to the sampling of air, separating updrafts and downdrafts on the condition of the sign of the vertical wind velocity. The mass flow rate of physical air samples needs to be proportional to the magnitude of the vertical wind velocity and controlled at 10 Hz or above to resolve flux-relevant turbulence scales. For conserved scalars, the net flux can then be determined from the difference in scalar concentration between the accumulated updraft and downdraft samples, respectively, over a certain flux integration interval, e.g., 30 minutes.

The idea of eddy accumulation (EA) goes back to early considerations by Desjardins (1972) who proposed the method for physically sampling trace gas fluxes. He reported a first experiment of conditionally sampling temperature and deriving sensible heat flux through mathematical accumulation (Desjardins, 1977). I use the term 'true eddy accumulation' rather than just 'eddy accumulation' to refer to the original formulation of eddy accumulation (Desjardins, 1977; Hicks and McMillen, 1984), specifically with vertical wind proportional air sampling, as opposed to later derivatives of eddy accumulation such as 'relaxed eddy accumulation', which is subject to constant mass flow and further limitations (see Sec. 1.1.2).

Literature on true eddy accumulation is sparse with just over a dozen published studies. Very few studies performed actual flux measurements. Desjardins (1977), Speer et al. (1985), Neumann et al. (1989), Beier (1991), and Komori et al. (2004) presented early prototypes of true eddy accumulators and disjunct true eddy accumulators (Rinne et al., 2000). Others conducted simulations (Hicks and McMillen, 1984; Businger and Oncley, 1990), contributed technology (Buckley et al., 1988) and reviews (Businger, 1986; Speer et al., 1986; Hicks et al., 1986). However, the practical implementation has long been difficult, particularly the accurate and dynamic control of mass flow rates. None of the experiments above produced significant long-term data sets. Correlation of TEA fluxes with EC fluxes was generally relatively low with coefficients of correlation of, e.g., $R^2 = 0.07$ (Speer et al., 1985), $R^2 = 0.41$ (Neumann et al., 1989), and $R^2 = 0.64$ (Komori et al., 2004)). Until today there is no TEA instrument commercially available.

Recently the author of the current study has successfully performed a series of TEA flux experiments using a new and fully digital approach to dynamic and fast mass flow control and real-time processing of wind data. Further advances concerned TEA flux corrections and TEA simulations. Those experiments yielded a tight correlation between TEA and EC flux measurements with coefficients of regression of up to $R^2 = 0.96$, exceeding $R^2$ values from any of the above cited literature. The current work presents an initial TEA and EC inter-comparison experiment performed over short vegetation during Spring 2015 in more detail.

The concept of the TEA sampling scheme is illustrated in Fig. 1. The true vertical wind velocity (top panel, black line) is sampled at a frequency of, e.g., 10 Hz (top panel, blue dots) using an ultrasonic anemometer. Likewise, the air sampling device samples the true atmospheric time series of the scalar, e.g., $CO_2$, (center panel, black line) at the same time resolution of 10 Hz (center panel, blue dots). The time variable flow rates at which samples are being accumulated are shown in the bottom panel. Separate accumulation of updrafts (red lines) and downdrafts (orange lines) are distinguished.





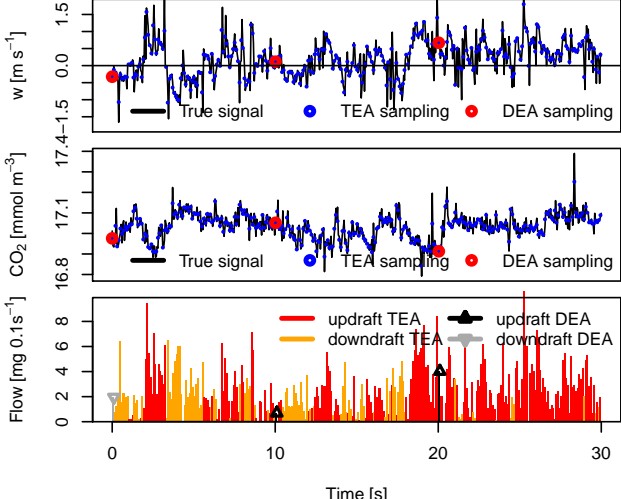

**Figure 1.** True eddy accumulation and disjunct eddy accumulation sampling scheme. Vertical wind, $w$, (top panel), scalar density, $CO_2$, (center panel) and vertical wind proportional mass flow rate (bottom panel). Black solid lines indicate the continuous true atmospheric signal. Sampling resolution of TEA and DEA are 10 Hz and 10 s, respectively. Note that active sampling time for DEA is only 1% of the sampling time for TEA.

Following whole air sampling, the atmospheric constituent of interest can be trapped in a number of ways. Constituents can be accumulated as whole air samples in bags, absorbed in gas washing reservoirs, adsorbed on to chemicals using cartridges, continuously sampled with denuders, trapped as reaction products with chemicals or retained using mechanical filters.

The true eddy accumulation principle is not limited to passive trace gases. Here, the author suggests that the TEA method

has the potential to measure fluxes of dust, pollen, bacteria, fungi and other biological material carrying physical, chemical and genetic information. The latter materials can be accumulated on appropriate filter media.

True eddy accumulation has a number of advantages over other methods. Sample accumulation over the duration of typical flux averaging intervals of 30 to 60 minutes allows for the use of slow-response gas analyzers. The key advantage of TEA over EC is the applicability to a much wider range of atmospheric constituents assuming that slow-response analyzers are more

readily available than fast-response analyzers and better accuracy can be obtained through signal averaging.

The key advantage of TEA over other variants of eddy accumulation, i.e., relaxed eddy accumulation or hyperbolic relaxed eddy accumulation, is that true eddy accumulation is the only direct method in the family of accumulation methods. As a direct method it does not require the use of proxies (other scalars) and coefficients like the $\beta$-coefficient in relaxed eddy accumulation and therefore does not depend on scalar similarity (Ruppert et al., 2006). This property of a direct measurement

method is essential for quantifying fluxes of constituents which cannot be measured by other means (e.g., the EC method). Scalar similarity of the fluxes of the constituent of interest and the proxy cannot be assessed without first quantifying both fluxes themselves. The direct TEA method is independent from prior knowledge.





Another advantage over other types of eddy accumulation (relaxed eddy accumulation or hyperbolic relaxed eddy accumulation) or any type of disjunct eddy sampling (e.g., the disjunct eddy covariance method or the disjunct eddy accumulation method) is the continuous sampling of the air by the TEA method such that the signal is recovered in its entirety. Continuous sampling avoids noise associated with disjunct sampling (Lenschow et al., 1994). Likewise, omitting samples at times of small

vertical wind velocities, which is common practice in relaxed eddy accumulation, would effectively be disjunct sampling, trading in noise for the sake of higher concentration differences between accumulated updrafts and downdrafts.

The long averaging intervals further allow for repeated measurements of the same sample, improving precision. The by design constant trace gas concentration of the accumulated samples at the time of analysis and the typically long analysis integration times are best matched with low sample flow rates through the gas analyzer. Low flow rates result in low power

consumption and low pressure drop over system components. A low pressure drop is beneficial for the stability and accuracy of the gas analyzer's reading.

### 1.1.2 Relaxed eddy accumulation (REA)

Given the challenges associated with the original formulation of true eddy accumulation, Businger and Oncley (1990) proposed a modified version of eddy accumulation, today known as relaxed eddy accumulation (REA). REA is based on the concept of

15 flux-variance similarity. In order to relate the scalar flux to the variance of the vertical wind velocity, a proportionality factor, $\beta$, was introduced, so REA became an indirect method.

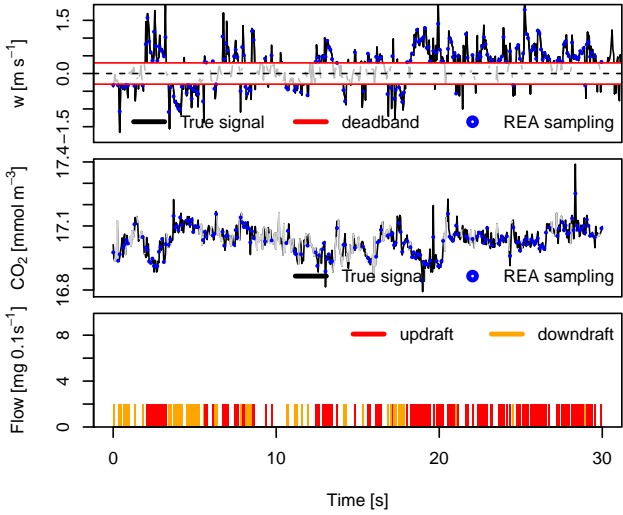

**Figure 2.** Relaxed eddy accumulation sampling scheme. Vertical wind, $w$, (top panel), scalar density, $CO_2$, (center panel) and mass flow rate (bottom panel). Black solid lines indicate the continuous true atmospheric signal. Sampling resolution of REA is 10 Hz. A fraction of the $CO_2$ time series (gray lines, center panel) is not sampled by REA due to use of a vertical wind velocity "dead-band" for small velocities (thresholds indicated by red lines, top panel). This dead-band causes gaps in the otherwise constant mass flow rate (bottom panel).



The advantage of the relaxed eddy accumulation method is that air is sampled at a constant flow rate (Fig. 2, bottom panel). This meant that the dynamic high frequency modulation of flow rates as a function of the magnitude of the vertical wind velocity as in the TEA method was no longer required. REA still accumulates updrafts and downdraft separately controlled by the sign of the vertical wind velocity.

5   A second modification was introduced in REA: at times of small positive or negative vertical wind velocities no air samples are taken. This "dead-band" is illustrated in Fig. 2, top panel. The center panel shows the air sampling scheme: the true scalar time series, e.g., $CO_2$ density (black line), is sampled at a regular frequency of, e.g., 10 Hz (blue dots) if the vertical wind velocity (Fig. 2, top panel, black line), sampled at the same 10 Hz frequency (Fig. 2, top panel, blue dots), is larger than the thresholds defining the dead-band. A certain fraction of the scalar time series is thus omitted from sampling (Fig. 2, center panel, gray line).

The use of a dead-band has two advantages: the concentration difference between the updraft and downdraft accumulated samples increase (Pattey et al., 1993; Katul et al., 1996), improving the ratio of the flux signal to the noise of the gas analyzer. Secondly, use of a dead-band leads to less frequent switching between updraft and downdraft samples, which relaxes the need for fast-response valves to some degree and would reduce material wear. One disadvantage of the dead-band is the effectively disjunct sampling, which causes noise in the flux estimates (Lenschow et al., 1994). Another disadvantage is the impact of the dead-band on the flux itself of unknown magnitude, depending on the co-spectrum of scalar and vertical wind velocity.

The simplifications of the REA method relative to the TEA method, particularly the constant mass flow rate, have facilitated wide adoption of the REA method. More than 200 studies on REA flux measurements and simulations have been published since its description less than 30 years ago (Businger and Oncley, 1990). The significant number of REA studies suggests that there is a need for alternatives to the eddy covariance method for certain applications.

Despite being simpler to implement than TEA, REA has distinct disadvantages. Being an indirect method, the accuracy of REA remains critically dependent on the correct determination of an a priori unknown $\beta$-factor. $\beta$ varies with scalar and with atmospheric conditions. Typical $\beta$ values obtained from measurements and simulations (Wyngaard and Moeng, 1992; Businger and Oncley, 1990; Oncley et al., 1993; Pattey et al., 1993; Baker et al., 1992; Gao, 1995; Milne et al., 1999; Katul et al., 1996; Baker, 2000; Ammann and Meixner, 2002; Held et al., 2008) are around 0.55, but range from ca. 0.4 to ca. 0.7, introducing significant uncertainty of up to several tens of percent of the measured flux.

Scalar similarity between a constituent of interest and a suitable proxy for determination of the $\beta$-factor is often lacking (Ruppert et al., 2006; Cancelli et al., 2015). The alternative use of a constant $\beta$-factor leads to up to one order of magnitude lower accuracy of the estimated flux (Foken and Napo, 2008).

A variant of REA is hyperbolic relaxed eddy accumulation (HREA) (Shaw, 1985; Bowling et al., 1999). HREA maximizes concentration differences between accumulation reservoirs through use of hyperbolic dead-bands. Thus, HREA can resolve small fluxes such as stable isotope fluxes of $^{13}C$ and $^{18}O$ (Bowling et al., 1999; Wichura et al., 2000). However, HREA requires proxies similar to REA and omits about two thirds of total sampling time through aggressive use of dead-bands, resulting in an increase of noise and therefore uncertainty.





### 1.1.3 Disjunct eddy accumulation (DEA) and disjunct eddy covariance (DEC)

Disjunct eddy sampling (Rinne et al., 2000; Turnipseed et al., 2009) is based on considerations by Lenschow et al. (1994) on representing turbulent time series by temporal subsamples. Disjunct eddy covariance (DEC) takes very short grab samples (ca. 0.1 s), followed by a pause (e.g., 5 to 60 s) for gas analysis with relatively slow instruments. Similarly, disjunct eddy

accumulation can be used to obtain short grab samples at a mass flow rate proportional to the magnitude of vertical wind velocity when continuous dynamic mass flow control can not be performed.

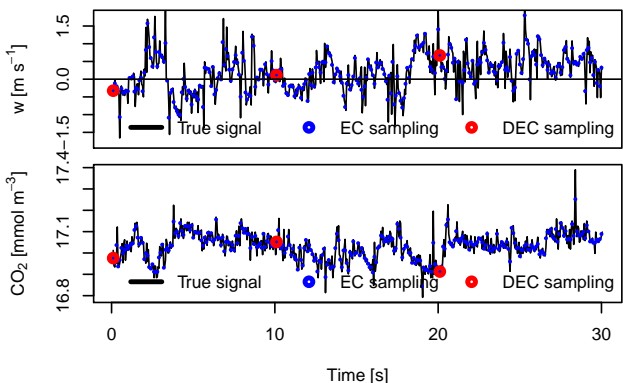

**Figure 3.** Eddy covariance and disjunct eddy covariance sampling scheme. Vertical wind, $w$, (top panel), scalar density, $CO_2$, (bottom panel). Black solid lines indicate the continuous true atmospheric signal. Sampling resolution of EC and DEC are 10 Hz and 10 s, respectively. Note that active sampling time for DEC is only 1% of the sampling time for EC.

The disjunct sampling principle is illustrated in Fig. 1 for the DEA method and in Fig. 3 for the DEC method. Comparing the few disjunct samplings at a resolution of 10 s of the DEA method (flow rate indicated by black vertical lines in the bottom panel of Fig. 1 at times 0, 10, and 20 s) relative to the continuous flow rate of the TEA method (red and orange vertical lines in

the bottom panel of Fig. 1) illustrates the small fraction of the total time series being actually sampled by DEA.

While allowing more time for the analysis of the chemical species, disjunct sampling introduces noise in the fluxes. Turnipseed et al. (2009) found an additional uncertainty of $\pm 30\%$ due to disjunct sampling and estimated the overall uncertainty of their DEA measurements to $\pm 40\%$.

### 1.1.4 Challenges of eddy accumulation

There are a number of challenges associated with eddy accumulation flux measurements (see also Hicks and McMillen (1984)). The first two listed below are specific to the TEA and DEA methods. The others are common to all eddy accumulation methods.

1. Mass flow control: The air sampling, i.e., the separation of updrafts and downdrafts as well as the response of the vertical wind velocity proportional mass flow control, needs to be sufficiently fast (10 Hz to 20 Hz) and dynamic to resolve the relevant turbulent fluctuations. Further, the mass flow control needs to be accurate even under dynamically changing flow



rate conditions despite the compressibility of air. Finally, the dynamic range of the mass flow control, i.e., the ratio of the largest to the smallest accurately controllable mass flow needs to be on the order of 100 or higher to limit flux errors (Hicks and McMillen, 1984). No commercially or otherwise readily available technology for fast, dynamic, and accurate control of mass flow rates exists or has been demonstrated to perform well in TEA.

2. Density fluctuations effects: Density fluctuations due to heat and water vapor transfer affect the flux of the scalar of interest. Corresponding corrections specific to the TEA method have been unavailable.

3. Spectra and co-spectra: No turbulence spectrum of the scalar nor the co-spectrum of the scalar and the vertical wind velocity can be obtained from the accumulated samples as they are mixed and time-resolved analysis is therefore not possible. Spectral information on the wind is of course available as in any other method using a fast-response ultrasonic anemometer.

4. Coordinate rotation (see further details in Sec. 2.6): Sampling decisions need to be performed in real-time and they are definitive, i.e., they cannot be modified in post-processing. This is an important difference to the EC method. The separation of updrafts and downdrafts depends on the definition of vertical wind velocity, $w$. The mean $w$ over the averaging period needs to be zero (see Sec. 2, Eq. 6). One way to minimize mean $w$ is to align the coordinate system of the wind measurements to the mean stream lines through coordinate rotation (Wilczak et al., 2001).

   In eddy accumulation, the coordinate rotation needs to be performed in pre-processing to be available in real-time. Coordinate rotation and any other operation attempting to nullify mean $w$ over the flux averaging interval would require knowledge of $w$ over the entire interval, including future observations. However, only past and present data are available in real-time to approximate coordinate rotation and perform sampling decisions based on the sign and magnitude of $w$. Remaining non-zero mean $w$ causes flux bias.

5. Decorrelation through sensor separation: Spatial separation of the gas sampling inlets and the wind sensing volume causes a time-lag between the wind measurement and the time for obtaining the corresponding air sample. Not accounting for time-lags leads to decorrelation of wind and scalar and therefore flux loss. Contrary to the EC method, where time-lags can be detected through covariance maximization and corrected for, in eddy accumulation such post-processing is not possible because high-frequency scalar time series are not obtained. Therefore the wind measurement and the air inlet need to be collocated as close as possible.

6. Analyzer sensitivity: Trace gas concentration differences between reservoirs might be too small to be resolved by a given gas analyzers.

7. Reliability: Eddy accumulation systems are mechanically and electronically complex machines. Particularly moving parts pose the risk of failure. Careful design is required for robust implementations and unattended long-term deployments.

We address above challenges in the following ways:



1. A new type of digital and highly dynamic mass flow controller was deployed. The technology previously developed by the author has fast and dynamic response sufficient to resolve relevant turbulent scales at 10 Hz and above. The design accounts for the compressibility of air in dynamic sampling.

2. Flux corrections are being developed by the author and will be reported separately.

3. As above.

4. Non-zero mean vertical wind velocities were minimized through real-time coordinate rotation with continuous near real-time updates of the rotation coefficients as well as a procedure to minimize remaining bias in the mean vertical wind velocity (see Sec. 2). Further, a correction accounting for volume mismatch between updraft and downdraft reservoirs due to non-zero mean vertical wind velocities (Turnipseed et al., 2009) has been applied (see Sec. 2).

5. Spatial separation of the wind sensing volume and the point of air sampling can be minimized through integration of the air inlet inside the wind sensing volume of the sonic anemometer. A certain degree of time lag between the wind signal and the scalar sampling will ultimately remain as long as the wind is sensed over a measurement volume rather than at a point and with discrete finite time resolution rather than truly instantly.

6. Performance analysis of a typical infrared gas analyzer model for the measurement of $CO_2$ gave satisfactory results in terms of the resolution but revealed limited stability. Subsequent work by the author with current laser spectrometers gave superior results due to their improved stability. Details on the latter work will be reported separately.

7. Suggestions towards a robust design of an eddy accumulator are given in the conclusions of the current study.

## 1.2 Objectives

Out of all the methods discussed above, the true eddy accumulation method is the only alternative to the eddy covariance method for directly measuring the physical flux. Every effort should be made towards mastering the dynamic mass flow control necessary for direct TEA as well as addressing the other challenges listed above.

It is the objective of this work to deliver the proof-of-concept of true eddy accumulation trace gas flux measurements based on dynamic and accurate mass flow control proportional to vertical wind velocity and based on fully digital and real-time signal processing.

## 2 Materials and Methods

### 2.1 Experimental design

The experimental design comprises three elements: novel true eddy accumulation flux measurements of $CO_2$ and water vapor, conventional eddy covariance flux measurements of $CO_2$ and water vapor, and supporting meteorological measurements. All measurements were performed side-by-side. This allowed for evaluating the performance of the new TEA system in its ability to measure turbulent fluxes of $CO_2$ by relating the observed fluxes to meteorological drivers and by comparing the TEA $CO_2$



flux measurements to conventional EC $CO_2$ flux measurements. This section provides details on the measurement site, the meteorological measurements, the TEA method, technical implementation, and flux computation as well as information on the reference flux measurements by EC.

## 2.2 Experimental site

Flux measurements were performed at a grassland experimental field site of the University of Goettingen, Germany. A flux tower was installed at an altitude of 230 m above sea level on a flat area of about 50 by 50 m situated on a South-Southeast facing hill with a slope angle of $5°$ and length of 800 m. Vegetation height of the grass was 0.2 m during the experiment. Vegetation further comprised patches of bushes and trees with a minimum distance from the flux tower of 50 m (West of the tower).

## 2.3 Experimental period

The TEA flux measurements presented in this study were conducted from April 4 to 10, 2015. After a cold and wet month of March, this period was characterized by increasing physiological activity of the grasses due to increasing light availability, increasing temperatures during the day, less frequent frost events and increasing $CO_2$ fluxes.

Prior to the flux experiment, the TEA instrument and method was further developed and tested in the field with continuous operation starting on March 5, 2015. The TEA deployment continued after April 10 until June 17, 2015. However, the frequent charging and discharging of the air sampling bags lead to material fatigue and progressive leakage. Therefore, no meaningful flux measurements are available after April 10, 2015. The period from April 10 to June 17, 2015, was used for testing different kinds of air bags and for further developing the TEA method. Altogether, the TEA air sampling system in its current form was in continuous operation from March 5 to June 17, 2015, corresponding to more than 5000 30-min TEA flux sampling intervals.

## 2.4 Instrumentation

The instrumentation used for meteorological measurements, TEA flux measurements, and EC flux measurements and the respective variables measured are listed in Tab. 1.

Meteorological variables (Tab. 1) were logged using a DL16 data logger (Adolf Thies GmbH & Co. KG Goettingen, Germany). All raw data needed for TEA and EC flux measurements, including the sonic anemometer data, and data from the two infrared gas analyzers LI-6262 and LI-7500 were synchronized and logged on the central TEA controller. Using a mobile network link, raw data were continuously mirrored to a central server for archival and flux processing.

## 2.5 Meteorological measurements

The following set of meteorological variables was measured on site to support the computation and interpretation of turbulent fluxes: global radiation, photosynthetically active radiation (PAR), net radiation, air temperature at 2 m a.g., air pressure,



**Table 1.** Instrumentation used for turbulent flux and meteorological measurements. Manufacturer key: Gill Instruments (Lymington, UK), Li-COR Environmental Inc. (Lincoln, Nebraska, USA), Kipp & Zonen B.V. (Delft, The Netherlands), Bosch Sensortec (Stuttgart, Germany), Adolf Thies GmbH & Co. KG (Goettingen, Germany), Imko Mikromodultechnik GmbH (Ettlingen, Germany), Hukseflux Thermal Sensors B.V. (Delft, The Netherlands).

| Variable | Sensor | Manuf. | Method | Freq. |
|---|---|---|---|---|
| Wind u,v,w | R3 | Gill | TEA, EC | 10 Hz |
| Sonic temp. Ts | R3 | Gill | TEA, EC | 10 Hz |
| $CO_2$ density | LI-7500 | Li-COR | EC | 10 Hz |
| $H_2O$ density | LI-7500 | Li-COR | EC | 10 Hz |
| $CO_2$ density | LI-6262 | Li-COR | TEA | 1 Hz |
| $H_2O$ density | LI-6262 | Li-COR | TEA | 1 Hz |
| Air pressure | BME280 | Bosch | TEA | 1 Hz |
| Air temperature | BME280 | Bosch | TEA | 1 Hz |
| Global radiation | CMP3 | Kipp | Meteo | 10 min |
| Photon flux density | PAR sensor | Thies | Meteo | 10 min |
| Net radiation | NR Lite | Kipp | Meteo | 10 min |
| Air pressure | DL16 internal | Thies | Meteo | 10 min |
| Air temperature | Galtec | Thies | Meteo | 10 min |
| Precipitation | Tipping bucket | Thies | Meteo | 10 min |
| Wind velocity | Cup anemometer | Thies | Meteo | 10 min |
| Wind direction | Wind vane | Thies | Meteo | 10 min |
| Soil temperature | Trime Pico 32 | Imko | Meteo | 10 min |
| Soil moisture | Trime Pico 32 | Imko | Meteo | 10 min |
| Soil heat flux | HFP01 | Huksef. | Meteo | 10 min |



relative humidity at 2 m a.g., precipitation, wind velocity at 2 m a.g., wind direction at 2 m a.g., soil temperature at 0.3 m below ground (three probes) and soil moisture at 0.3 m below ground (three probes) and soil heat flux.

## 2.6 Coordinate systems and net ecosystem exchange

If the trace gas source and sink strength of the ecosystem is of interest, as is typically the case when investigating biological, physiological or biogeochemical processes or deriving trace gas budgets, then the total flux in and out of the ecosystem needs to be determined. For the exchange of the ecosystem with the atmosphere, the concept of a virtual control volume is often used. Net ecosystem exchange ($NEE$), i.e., the net flux across the surfaces of this control volume, can be written as (e.g., Aubinet et al., 2003; Siebicke et al., 2012):

$$NEE = \underbrace{\frac{1}{V_m} \int_0^h \left( \frac{\partial \overline{c}}{\partial t} \right) dz}_{\text{I}} + \underbrace{\frac{1}{V_m} \left( \overline{w'c'} \right)_h}_{\text{II}}$$

$$+ \frac{1}{V_m} \int_0^h \left( \underbrace{\overline{w}(z) \frac{\partial \overline{c}}{\partial z}}_{\text{IIIa}} + \underbrace{\overline{c}(z) \frac{\partial \overline{w}}{\partial z}}_{\text{IIIb}} \right) dz$$

$$+ \underbrace{\frac{1}{V_m} \int_0^h \left( \overline{u}(z) \frac{\partial \overline{c}}{\partial x} + \overline{v}(z) \frac{\partial \overline{c}}{\partial y} \right) dz}_{\text{IV}}$$

$$+ \underbrace{\frac{1}{V_m} \int_0^h \left( \frac{\partial \left( \overline{u'c'} \right)}{\partial x} + \frac{\partial \left( \overline{v'c'} \right)}{\partial y} \right) dz}_{\text{V}} \tag{1}$$

with the molar volume of dry air $V_m$, $CO_2$ concentration $c$, time $t$, horizontal distances $x$ and $y$, vertical distance above ground $z$, height of the control volume $h$, horizontal wind velocity $u$ along the $x$-direction, horizontal wind velocity $v$ along the $y$-direction and vertical wind velocity $w$ along the $z$-direction. Overbars denote temporal means and primes denote the temporal fluctuations relative to the temporal mean.

The terms on the right hand side of Eq. 1 are the change of storage (I), the vertical turbulent flux (II), vertical advection (IIIa), vertical mass flow from the surface e.g. due to evaporation (IIIb) according to Webb et al. (1980), horizontal advection (IV), and flux divergence (V). The form of $NEE$ presented in Eq. 1 excludes the horizontal variation of the vertical turbulent flux and the horizontal variation of vertical advection. Most flux measurements typically only determine the vertical turbulent flux density (term II) and sometimes the storage flux density (term I), neglecting the remaining terms due to a lack of spatially distributed information.





The choice of the reference coordinate system (Finnigan, 2004) is important for the attribution of the total flux to its components (Eq. 1) and therefore for the interpretation of turbulent flux density measurements relative in their ability to approximate the net ecosystem exchange. If NEE is to be assessed, and available flux observations are restricted to the turbulent vertical flux density at a single location above the ecosystem, a reference coordinate system is needed which minimizes the remaining flux terms. Sun et al. (2007, Tab. 1) summarize coordinate systems. In stream line coordinates (Finnigan, 2004; Sun et al., 2007), the long-term flow is tangential to long-term stream lines. This means that, in stream line coordinates, the velocity normal to the stream lines becomes zero, implying that the long-term vertical advection vanishes. There are various methods for coordinate rotation (Finnigan et al., 2003), i.e. the transformation of the wind measurements from the coordinate reference frame of the sonic anemometer to the coordinate system of the flux measurements, also known as tilt correction (Tanner and Thurtell, 1969; Hyson et al., 1977; Kaimal and Finnigan, 1994; McMillen, 1988; Paw U et al., 2000; Wilczak et al., 2001).

Over flat terrain or planar terrain with a uniform slope, the mean stream lines close to the surface approximately follow the terrain surface. The planar fit method (Paw U et al., 2000; Wilczak et al., 2001) is often used to obtain long-term stream line coordinates. In contrast to the double rotation method (Kaimal and Finnigan, 1994), which nullifies the mean vertical wind velocity $\overline{w}$ of the flux integration interval, planar fit rotated $\overline{w}$, using the original formulation of Wilczak et al. (2001), is typically small but not zero. Even the long-term $\overline{w}$ only becomes zero if mean stream lines are planar and there is no instrumental bias in measurements of $w$. Non-zero $\overline{w}$ would imply existence of a vertical advection term proportional to $\overline{w}$ in the presence of vertical trace gas concentration gradients, which for $CO_2$ typically exist close to the surface.

A variant to the planar fit method proposed by Van Dijk et al. (2004) removes velocity bias relative to the flux integration interval, addressing instrument offsets. This procedure can lead to local misalignment of streamlines for non-planar mean flow fields. Nullifying $\overline{w}$ over the flux integration period would formally remove vertical advection terms from the flux budget equation (Eq. (1)). However, also this procedure would still ignore the effect of misalignment of the reference coordinate system and the stream lines over the flux integration interval on the vertical turbulent flux. The mismatched length and timing of the planar fit period relative to the shorter individual flux integration intervals acts as a high pass filter and results in loss of low frequency flux contributions and in unresolved distortion of co-spectra of the shorter flux intervals (Finnigan et al., 2003).

On a related matter, Siebicke et al. (2012) performed an explicit treatment of the length and timing of the reference period for planar fit rotation. They demonstrated changes of up to 50% of advective $CO_2$ fluxes over forest depending on the window size of a new serial planar fit approach.

Over complex non-planar terrain, the mean stream lines are not tangential to a plane. Even over planar surfaces, stream lines further away from the surface may not be tangential to the terrain surface due to vertical velocity divergence (Sun et al., 2007). For curved stream lines, other terms of the mass flow equation (Finnigan, 2004), in addition to vertical turbulent flux and vertical advection, become important. For curved stream lines, horizontal trace gas advection is not proportional to the gradient of trace gas concentrations along the stream lines.

Several authors have suggested variants to the planar fit method to account for steep slopes, where buoyancy forces are no longer normal to the terrain surface (Oldroyd et al., 2016); for obstructed flow Griessbaum and Schmidt (2009); and for complex topography, where $\overline{w}$ becomes a function of wind azimuth angle. Consequently, several studies apply the planar fit





rotation separately for different wind direction sectors (Foken and Napo, 2008; Yuan et al., 2011, and others). However, this introduces discontinuities in $\overline{w}$ at wind direction sector limits and in the definition of the reference coordinate system.

A more general approach avoiding directional discontinuities would be the method of fitting a surface (hereafter referred to as "surface fit") rather than a plane (the author, unpublished work), where the curvature of the surface adapts to long-term

stream lines as a function of one or more parameters, i.e. wind direction (one-dimensional surface fit approach) and optionally other variables, such as horizontal wind velocity (multi-dimensional surface fit approach). Siebicke et al. (2012) showed that the effect of atmospheric stratification, friction velocity, stationarity and integral turbulence characteristics (Foken et al., 2004) on sectoral planar fit rotation was small relative to the wind direction effect over a forest. A surface fit related approach has recently been proposed by Ross and Grant (2015), who also suggest tilt correction as a continuous function of wind direction

(i.e. one dimension) instead of the relatively common discrete sectoral approach used currently.

In the presence of flow distortion due to obstacles, terrain features, towers and sensor mounts, or non-omnidirectional sonic anemometer designs (Li et al., 2013), distorted sectors need to be excluded from the definition of the coordinate system and subsequent flux derivations, unless distortions are characterized and corrected for (Van Dijk et al., 2004; Griessbaum and Schmidt, 2009, see also Sec. 2.7, current study).

All above considerations on coordinate rotation apply to both the EC and EA methods, respectively, including their derivatives. However, there is one conceptual difference: in EC, given high frequency observations of both the scalar and the wind and the possibility of flux post-processing, the length and time localization of the coordinate rotation reference period may comprehend all shorter flux integration intervals it applies to. On the contrary, in EA, in absence of high frequency scalar observations, any decision on the reference coordinate system becomes final on obtaining and mixing individual high frequency

air samples, precluding post-processing and any reconsideration of the coordinate system. In EA, the reference period defining the coordinate system necessarily cannot coincide and never fully overlap with the flux integration interval for any sample but the last one in the flux integration interval, if any. Due to this conceptual difference, the contribution of other flux terms, in particular vertical advection, may not be identical for EC and EA if not using the same reference period to align the coordinate system to the mean stream lines.

Non-vanishing mean vertical wind velocities $\overline{w}$ over the flux integration interval can nominally be removed in EC through subtraction of $\overline{w}$ from instant vertical velocities (Van Dijk et al., 2004) and in EA through application of the volume mismatch correction (Turnipseed et al., 2009), see Eq. 10 above. However, distortion of co-spectra (Finnigan et al., 2003) remains uncorrected.

We distinguish in evaluating the implications of discussed deviations of the real flow from any chosen ideal reference

conditions: (i) the case of deploying turbulent vertical flux measurements to estimate net ecosystem exchange, and (ii) the case of comparing the EA method and instruments side-by-side to the chosen reference method EC for assessing whether the EA method's physical air sampling principle produces comparable results to the mathematical computation of covariances for the EC method. This study is concerned with the latter case only. Most of the above issues of coordinate frames and the spatio-temporal variability of the flow field afflict both methods alike. Only the unavoidable differences in the application of the

coordinate rotations between the two methods, i.e. the non-matching rotation periods, need to be of concern when evaluating the





relative performance of the two turbulent flux observation methods. To eliminate this remaining difference, identical rotation procedures and planar fit reference periods need to be applied to both EA and EC, accepting the EA version as the reference.

## 2.7 Flow distortion and angle of attack correction

The physical structure of sonic anemometer probes distorts the air flow they intend to measure (Wyngaard, 1981), introduc-
ing systematic errors in flux measurements. Measurement errors, due to probe-induced flow distortion and self-sheltering of ultrasonic transducers, (Gash and Dolman, 2003), depend on the angle-of-attack (Kaimal and Finnigan, 1994), i.e. the angle between horizontal and the instantaneous wind vector. van der Molen et al. (2004) provided a wind tunnel calibration for anemometer models R2 and R3 (Gill Instruments Ltd., UK), updated by Nakai et al. (2006). The representativity of the wind tunnel calibrations for turbulent conditions in the field has been questioned (Högström and Smedman, 2004) and is still under
debate (Huq et al., 2017). Nakai and Shimoyama (2012) proposed an improved correction based on field measurements under turbulent conditions for the R3 and Windmaster models (Gill Instruments Ltd., UK). There is still no consensus on whether this correction should be applied, and care must be taken as the correction applies to certain instrument models (Gill Windmaster) and serial numbers only.

In the current study, which uses two R3-type anemometers (Gill Instruments Ltd., UK), we do not apply any angle of attack
correction because: (i) the applicability of the wind tunnel calibration (Nakai et al., 2006) may or may not be applicable; (ii) there is contrasting information on the applicability of the calibration under turbulent conditions (Nakai and Shimoyama, 2012), specifically to the R3 model (recommended for R3 by original authors but not according to later information from Gill Instruments, UK, and LI-COR Env., USA); (iii) no angle-of-attack correction was available in the current TEA system software at the time of the field experiment nor can the TEA flux measurements be post-processed to include the correction. For the
above reasons, no angle-of-attack correction was applied to the presented results of the current study, neither to TEA nor to EC fluxes. However, we did assess the impact of the angle-of-attack correction on EC $CO_2$ fluxes for the two sonic anemometers.

## 2.8 True eddy accumulation (TEA) flux measurements

### 2.8.1 TEA method

The true eddy accumulation method determines the flux, $\overline{wc}$, of a scalar (such as a trace gas) as the sum of the covariance, $\overline{w'c'}$
of the scalar, $c$, and the vertical wind velocity, $w$ and the product of the time averages of scalar and vertical wind velocity, $\overline{w}\,\overline{c}$, as

$$\overline{wc} = \overline{w'c'} + \overline{w}\,\overline{c} \qquad (2)$$

where over-bars denote time averages over the averaging period, $T_{avg}$, and primes denote fluctuations from the mean. Eq. 2 is analog to the eddy covariance (EC) method.





However, in contrast to EC, which requires high frequency observations of the scalar and the vertical wind velocity and mathematically deriving the covariance through post-processing, in the case of TEA, the separate sampling of the wind and scalar time series is replaced by physically collecting separate air samples of updrafts and downdrafts proportionally to the magnitude of the vertical wind velocity. The TEA flux over a given averaging period, $T_{avg}$, can thus be obtained as (Desjardins, 1977; Hicks and McMillen, 1984)

$$\overline{wc} = \frac{1}{T_{avg}} \int_0^{T_{avg}} (\delta^+ cw + \delta^- cw)dt \tag{3}$$

where $\delta^+ = 1$ when $w > \overline{w}$ and 0 when $w < \overline{w}$, and $\delta^- = 1$ when $w < \overline{w}$ and 0 when $w > \overline{w}$. The amount of air, $cw$, sampled per unit time, $dt$, contains the molar fraction of the scalar of interest, $c$.

Assuming ideal conditions such that the mean vertical wind velocity over the integration period, $\overline{w}$, was zero, the mean term $\overline{w}\,\overline{c}$ becomes zero and the scalar flux, $F_c$ becomes

$$F_c = \overline{w'c'} \tag{4}$$

in kinematic units of $\mathrm{m\,s^{-1}}$. Multiplying by moist air density $\rho$, we obtain the constituent mass flux, $F_c$, per unit area and unit time in units of $\mathrm{kg\,m^{-2}\,s^{-1}}$ as

$$F_c = \overline{w'c'}\overline{\rho} \tag{5}$$

Given $\overline{w} = 0$, the volume of the air samples accumulated in the updraft reservoir is identical to the volume of the downdraft reservoir given

$$w^+ + w^- = \overline{w} = 0 \tag{6}$$

where $w^+$ is vertical wind larger than $\overline{w}$ and $w^-$ is vertical wind smaller than $\overline{w}$.

A practical implementation of TEA then determines the scalar flux, $F_c$, as half of the difference between the mole fraction of the scalar in the updraft reservoir, $\overline{c^+}$, and the mole fraction of the scalar in the downdraft reservoir, $\overline{c^-}$, multiplied by the mean of the absolute value of vertical wind velocity, assuming $\overline{w} = 0$.

$$F_c = \frac{\overline{|w|}}{2}(\overline{c^+} - \overline{c^-})\overline{\rho} \tag{7}$$

in units of $(\mathrm{kg\,m^{-2}\,s^{-1}})$ or

$$F_c = \frac{\overline{|w|}}{2}(\overline{c^+} - \overline{c^-})\frac{1}{V_m} \tag{8}$$

in units of $(\mathrm{mol\,m^{-2}\,s^{-1}})$, with the molar volume of air, $V_m$ ($\mathrm{m^3\,mol^{-1}}$), according to the ideal gas law,

$$V_m = \frac{RT}{P} \tag{9}$$





with temperature, $T$, pressure, $P$, and the ideal gas constant, $R$.

In any practical flux measurement application the observed mean vertical wind velocity over the integration period is likely unequal to zero, i.e., $\overline{w} \neq 0$. In eddy covariance the assumption of $\overline{w} = 0$ is satisfied in post-processing once all observations of the entire integration period are available. This is commonly achieved by rotating the coordinate frame of the wind measure-

ments to minimize or even nullify $\overline{w}$ followed by subtraction of $\overline{w}$ from individual vertical wind velocity measurements, $w$, so $\overline{w}$ becomes zero.

On the contrary, for TEA, knowledge of the mean vertical wind velocity over the flux averaging period, $\overline{w}$, is required at any time throughout the averaging period, in order to be able to classify vertical winds as updrafts or downdrafts and accumulate air samples in the corresponding reservoirs. At any time during the averaging interval only the past and present vertical wind

measurements are known. Therefore, any attempt to obtain $\overline{w} = 0$ needs to rely in part on an estimate of the mean vertical wind velocity of the entire averaging period without knowledge of future observations from present through to the end of the averaging period. In practice, this situation can lead to $\overline{w} \neq 0$, resulting in unequal sample volumes accumulated over the averaging period in the updraft and downdraft reservoirs.

Following Turnipseed et al. (2009) the flux needs to be corrected by a term accounting for the mismatch between the volume

of accumulated updrafts, $V^+$, and downdrafts, $V^-$, respectively, for $V^+ \neq V^-$. The flux due to a mismatch of volumes $V^+$ and $V^-$ is

$$F_{c,\text{volume\_mismatch}} = \overline{w} \left( \frac{(\overline{c^+} + \overline{c^-})}{2} - \overline{{}^v c} \right) \frac{1}{V_m} \tag{10}$$

where the volume mismatch correction term is the difference between the unweighted mean density of the reservoirs, $(\overline{c^+} + \overline{c^-})/2$ and the volume weighted mean density,

$$\overline{{}^v c} = \frac{(\overline{c^+} V^+ + \overline{c^-} V^-)/2}{(V^+ + V^-)/2} \tag{11}$$

weighted by the updraft and downdraft volumes, $V^+$ and $V^-$, respectively.

Inserting Eq. (10) in Eq. (8) yields the volume mismatch corrected TEA flux,

$$F_c = \left( \frac{\overline{|w|}}{2} (\overline{c^+} - \overline{c^-}) + \overline{w} \left( \frac{(\overline{c^+} + \overline{c^-})}{2} - \overline{{}^v c} \right) \right) \frac{1}{V_m} \tag{12}$$

In practical applications, the instant sampling volume per unit time, $V_i$, is related to instant vertical wind velocity, $w_i$, through

a proportionality factor, $k$, as

$$V_i = k \, |w_i| \tag{13}$$

### 2.8.2   Correction of trace gas mole fractions for the effects of water vapor

When measuring trace gases such as $CO_2$ in moist air with infrared gas analyzers, two corrections are required to remove the effect of water vapor on the measurement of the mole fraction of the trace gas. The first correction accounts for pressure





broadening due to the presence of water vapor. This is known as the "pressure broadening correction" (Licor, 1996, Li-COR LI-6262 manual, Sec. 3.5, pp. 25, Eq. 3-30).

The second correction accounts for the dilution of the trace gas by water vapor in the sample. This correction is known as "dilution correction" and is required to convert wet mole fraction, $c_{wet}$, to dry mole fraction, $c_{dry}$. Dry mole fraction is needed

for calculating the trace gas flux, $F_c$, based on Eq. (5). The dry mole fraction is obtained from wet mole fraction following the instructions for the infrared gas analyzer (Licor, 1996)

$$c_s^{wr} = c_s^{ws} \left( \frac{1 - X_{w,ref}/1000}{1 - X_w/1000} \right) \tag{14}$$

where $X_w$ is the mole fraction of water vapor in the sample cell, $X_{w,ref}$ is the water vapor mole fraction in the reference cell, $c_s^{ws}$ is the actual mole fraction of the trace gas in the sample cell diluted by $X_w$, and $c_s^{wr}$ is the equivalent sample cell $CO_2$

mole fraction if it were diluted by $X_{w,ref}$.

### 2.8.3 TEA instrumentation and technical implementation

The TEA instrumentation used in this study was developed with particular attention to accurate and dynamic sampling of air and to real-time processing of wind data. The system was further designed to minimize time lags and jitter in wind data processing and in air sampling and to minimize dead-volumes in the gas sampling system.

Vertical and horizontal wind velocities and the sonic temperature were measured using an ultrasonic anemometer R3 (Gill Instruments Ltd.), the same type which was also used for the side-by-side eddy covariance reference flux measurements. Wind velocity data were logged at a 10 Hz frequency.

Instant observations of vertical wind velocity were subject to real-time coordinate rotation to align the coordinate system of the sonic anemometer with the mean stream lines prior to controlling the sampling of air into updraft or downdraft reservoirs.

To the author's knowledge, this study is the first eddy accumulation study to deploy a real-time coordinate rotation based on the planar fit rotation (Wilczak et al., 2001). A moving window of one day was used to estimate planar fit rotation coefficients with an update frequency of the rotation coefficients of 30 minutes. The coefficients were then applied to rotate the instant raw wind measurements, $w_i$, ten times per second.

To minimize $\overline{w}$ over the flux averaging period the following procedure was applied: the mean vertical wind velocity of the

current accumulation interval was approximated by the mean rotated vertical wind velocity over the most recent samples over a period with length equal to the length of the accumulation period, in this case 30 minutes. This estimate of mean vertical wind velocity was updated every two minutes and subtracted from every instantaneous vertical wind velocity measurement after coordinate rotation, i.e., ten times per second.

The decision on updrafts and downdrafts was based on the sign of rotated $w_i$. Sample volumes where computed following

Equation (13). $k$ was determined such that instant flow rates would not exceed the maximum possible flow rate of $3 \, \mathrm{l \, min}^{-1}$ with a probability of 99% based on absolute wind data in the period from 30 min ago to present. The proportionality factor, $k$, which was based on the 99%-quantile multiplied by a factor of two, was updated every 30 min.





Air inlets were collocated with the wind measurement and positioned 20 cm below the center of the sonic anemometer. Two separate air sampling lines were used, one for obtaining the updraft samples and one for the downdraft samples, respectively. In contrast to many previous eddy accumulation studies, which have used a single air inlet and a 3/2-way valve to direct the samples towards the appropriate updraft or downdraft reservoir, the current design with two separate sampling lines avoids any

undesired mixing of updrafts with downdrafts in the system.

The intake of air was controlled by fast response mass flow controllers with a dynamic response resolving turbulent eddies at 10 Hz. The mass flow controllers developed by the author were calibrated using conventional thermal mass flow controllers (Voegtlin ready smart series). The accuracy of the new dynamic mass flow controllers was equal to or better than 0.3%, which corresponds to the accuracy of the conventional mass flow controller model used for calibration. Air sampling lines were made

of Teflon with a 6 mm outer and 4 mm inner diameter and a length of 5 m between intake and accumulation reservoirs.

The air was filtered before entering the pumps and the bag reservoirs using PTFE membrane Gelman Acro Disc filters with a 50 mm diameter and a 2 μm pore size. Another filter was placed directly upstream of the gas analyzer.

At any time, one of the air sampling lines was active, with the selection of the line depending on the sign of the vertical wind velocity. The wind was measured at a frequency of 10 Hz using the sonic anemometer. With every new reading of vertical wind

velocity, i.e., every 100 ms, the selection of the active inlet (updraft or downdraft) was updated depending on the sign of the vertical wind velocity and an air sample obtained with mass proportional to vertical wind velocity.

In contrast to common practice in many relaxed eddy accumulation studies, which typically define a minimum vertical wind velocity for air sampling, in the current design, air samples were obtained for all magnitudes of vertical wind velocity (except for the 0.5% most positive and most negative values of $w$, respectively, as per the definition of $k$ above).

Air samples were collected using two variable speed separate brush-less DC membrane pumps (KNF Neuberger GmbH, Germany) with a maximum flow rate of 3 l min$^{-1}$, each, feeding air into the bag reservoirs at flow rates between 0 and 3 l min$^{-1}$.

Air was collected in lab grade, chemically inert Alumini® air sample bags (Westphalen AG, Germany) with a volume of 28 l. The composite wall of the bags was made of (from outside to inside) Polyethylene terephthalate (PET), Polyethylene (PE),

Aluminum (ALU), Oriented Polyamide (OPA), and Polyethylene (PE).

The layout of the TEA system is shown in Fig. 4. The system was designed for continuous operation, with continuous sampling of air and continuous on-site gas analysis. Air was collected in bags over periods of 30 min. Subsequently the air was analyzed over the following 30 min periods. A total of four bags was used at any time. A set of two bags (marked with "A" in Fig. 4) was charged with samples over 30 min, with one bag accumulating samples corresponding to updrafts (marked

"updraft") and one bag accumulating samples corresponding to downdrafts (marked "downdraft"). A second set of two bags (marked with "B" in Fig. 4), which contained air samples from the previous 30 min interval, was discharged and analyzed in parallel to the filling of the first set of bags. Every 30 min the two bag sets "A" and "B" would swap their function from being charged to being discharged. Note that data analysis revealed a leak in one of the bag sets, so that the data of every other half hour had to be discarded.



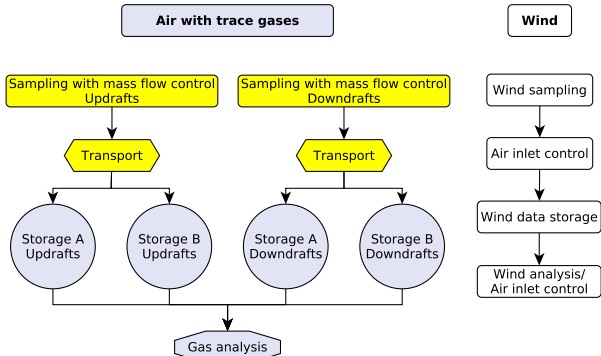

**Figure 4.** Schematic functional flowchart of true eddy accumulation system. Sampling of air with atmospheric constituents (scalars) shown on the left, sampling of wind vector shown on the right. Updrafts and downdrafts were sampled and stored via separate lines. One single analyzer was used with the sample supply alternating between the updraft and downdraft reservoirs every 150 s (see Fig. 5). During a particular 30 min period, bag set A was being filled with updraft and downdraft samples in the updraft and downdraft bags, respectively. At the same time bag set B was being analyzed and discharged, with the analysis alternating between updrafts and downdrafts. Every 30 minutes, the operation of bag sets A and B, either filling or discharging, would be swapped. The vertical wind velocity data (right) control the air sampling via instantaneous wind measurements and wind statistics.

The accumulated air was analyzed for molar density of $CO_2$ ($\mu mol\,mol-1$) and of water vapor ($mmol\,mol-1$) using a dual cell infrared gas analyzer, model LI-6262 (Li-COR Env. Inc, USA). Air samples were discharged by a membrane pump (KNF Neuberger, Germany) from the sample bags through the sample cell of the gas analyzer at a flow rate of $0.6\ l\ min^{-1}$. The sampling frequency of the gas analyzer was 1 Hz. The reference cell was purged by dry and $CO_2$-free zero gas obtained by circulating air through a scrubber filled with Soda lime and Dryrite desiccant. 3/2-way solenoid valves were used to select the appropriate gas bag for gas analysis.

During any 30 min period, the gas analysis alternated between sampling the updraft and downdraft reservoir, respectively (Fig. 5). Each bag was sampled for 150 s at a time. The first eight measurements periods of 150 s each were used for further analysis, resulting in four replicate measurements of the gas densities of the updraft reservoir and likewise four replicates of the downdraft reservoirs. Gas density measurements were tagged by the TEA controller with the respective active channel, either updraft or downdraft reservoir, for flux processing. The alternating sampling sequence lasted 1200 s. Over the remaining 600 s of the 30 min period, the remaining air was discharged from the bags to the atmosphere at a flow rate of $1.5\ l\ min^{-1}$ until depletion.

The fully automatic TEA system was controlled by an embedded computer, the 'TEA controller'. All sensor measurement data, including the wind and gas density measurements were synchronized, logged and processed on the same TEA controller.

The following raw data were logged for subsequent turbulent flux calculations: horizontal and vertical wind velocity components, $u, v, w$; and sonic temperature, $T_s$; $CO_2$ and $H_2O$ molar densities; analyzer cell temperature and cell pressure; ambient air temperature, $T$; and air pressure, $P$. Further data on the state of the TEA sampling system and the analysis were logged for



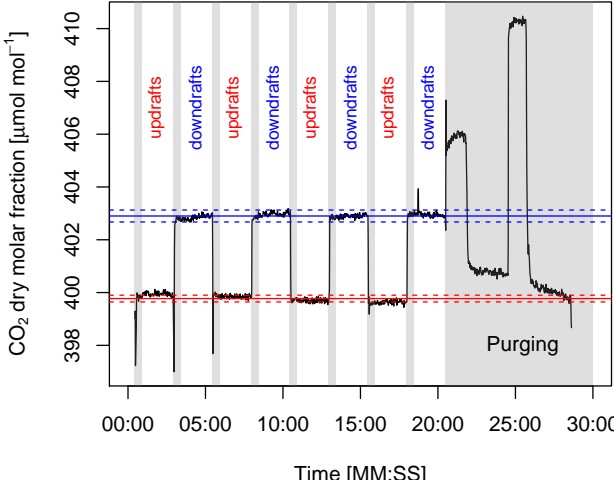

**Figure 5.** $CO_2$ dry molar fraction measured as a sequence alternating between updraft and downdraft reservoirs. First the updraft reservoir was measured for 150 s, then the downdraft reservoir for 150 s, discarding the initial 30 s of measurements from each block. This sequence was repeated four times. During the remaining 10 min of the 30 min period any remaining air in the reservoirs was purged and not used for analysis (gray shaded area). The mean of the despiked updraft and downdraft dry molar fractions are indicated by a red and blue solid line, respectively. The dashed lines indicate the mean $\pm$ 1 standard deviation. The start date of the time series is 2015-04-10 13:00:00 UTC.

attribution of the gas analyzer measurements to updraft and downdraft samples, for the selection of the bag sets and for system monitoring and quality control.

The energy-efficient TEA system of the current study consumed 15 W of electrical power (excluding the gas analyzer), about 10 W of which were used by the three pumps. The pumping power required for the current TEA system was two

orders of magnitude smaller than that of fast-flow closed-path eddy covariance systems using infrared gas analyzers or laser spectrometers of ca. 1 kW. The additional power consumption of typical current laser spectrometers would be on the order of 0.25 kW. The difference in pumping power between TEA and EC scales with the flow rate and sample cell vacuum of the EC application and is therefore even more important for most closed-path laser spectrometers than for infrared gas analyzers.

### 2.8.4 TEA flux computations

Turbulent fluxes of $CO_2$ were calculated from raw data of horizontal, $u, v$, and vertical, $w$, wind velocity components, and sonic temperature, $T_s$ (10 Hz), $CO_2$ dry mole fraction (converted from wet mole fraction) and $H_2O$ dry mole fraction (converted from wet mole fraction) (1 Hz), ambient air temperature, $T$ (10 min resolution), and pressure, $P$ (10 min resolution).

Raw gas density measurements were processed prior to flux computations in order to filter the data for noise and aggregate individual readings to single representations of the gas density, one for the updraft and one for the downdraft reservoir, during

any one 30 minute period. Blocks of 150 s (see Fig. 5) of measurements at 1 Hz were filtered and aggregated to 30 min values (see Sec. 3, Fig. 10 for results). The following statistically robust procedure was used:





- Conversion of raw voltage signals of $CO_2$ and $H_2O$ to physical units of $\mu mol\,mol^{-1}$ and correcting for the band broadening effect of pressure on $CO_2$ observations according to Licor (1996).

- Conversion of $CO_2$ wet mole fraction to dry mole fraction.

- Plausibility check of gas density data based on preset minimum and maximum values.

- Omission of the initial 30 s (dead-band filter) after switching channels to allow for purging shared gas handling components, i.e., valves, sample line, and gas analyzer sample cell.

- De-spiking of raw data (spike filter), using the function 'despike' from the R-package 'oce' (Kelley and Richards, 2017), replacing discarded values with the median of remaining values. The method identifies spikes with respect to a "reference" time series, and replaces these spikes with the reference value, i.e., here the median.

- Smoothing of the time series (smoothing filter) using the function 'loess' (Cleveland et al., 1992) from the R-package 'stats' (R Core Team, 2017). The function 'loess' fits a polynomial surface determined by one or more numerical predictors, using local fitting.

- Selection of stable readings (stationarity filter) by limiting maximum permissible gradients between individual samples for channel specific data blocks of 150 s (120 s remaining after dead-band filter), with a maximum permissible change
15
of $0.002\,\mu mol\,mol^{-1}\,s^{-1}$.

- Check for sufficient availability of data after filtering (availability filter): discard data block if less than 30 (out of of 120) values remain available after stationarity filter.

- Aggregation of the remaining filtered data per channel specific data block of 150 s using the median function.

- Aggregation of the four replicates per channel into a single value per 30 minutes as the weighted mean of the four
20
samples, weighted by the number of accepted raw measurements in each of the four replicates, separately for updrafts and downdrafts.

- To quantify precision of the $CO_2$ molar fraction measurements using the LI-6262 infrared gas analyzer, the minimum and maximum molar fractions over the four replicated samples were estimated, separately for updrafts and downdrafts and propagated into minimum and maximum flux estimates, respectively.

Fluxes were then calculated as follows:

- Plausibility check of wind data based on preset minimum and maximum values.

- De-spiking of raw wind data (Vickers and Mahrt, 1997).

- Computation of the mole fraction difference between updraft and downdraft reservoirs per 30 min period.

- Computation of uncorrected turbulent fluxes for 30 min intervals according to Eq. (8)

- Computation of turbulent fluxes for 30 min intervals, corrected for volume mismatch between updraft and downdraft reservoirs, according to Eq. (12)



### 2.9 Eddy covariance (EC) reference flux measurements

#### 2.9.1 EC instrumentation

A conventional EC system was set up for flux measurements of $CO_2$, sensible heat and latent heat. The EC setup served as a reference for the TEA flux measurements. Instruments used were a 3-dimensional sonic anemometer of type R3 (Gill Instruments Ltd, UK), and a infrared gas analyzer, type LI-7500 (Li-COR Env. Inc., USA). Wind and mole fraction data were recorded at a 20 Hz frequency at a height of 2.5 m above ground. The EC sensors were mounted side-by-side to a separate sonic anemometer and air inlet used for TEA. The two sonic anemometers were separated by a distance of 1 m. For quality assurance, in addition to above primary eddy covariance setup, we used the sonic anemometer of the eddy covariance in combination with the open-path fast response gas analyzer (IRGA) for an alternative eddy covariance flux estimate. The horizontal separation between the EC sonic and the IRGA was 0.35 m and the separation between the TEA sonic and the IRGA was 0.7 m.

#### 2.9.2 EC flux computations

Eddy covariance raw data were post-processed to obtain fluxes at a resolution of 30 minutes using the EddyPro® software (LI-COR Env. Inc., USA), version 5.0.0. The flux processing comprised the following steps:

- Statistical tests for raw data screening after Vickers and Mahrt (1997), including spike count and removal, amplitude resolution, drop-outs, absolute limits, skewness and kurtosis,
- De-trending of raw time series by block averaging,
- Compensation of time-lag between sonic anemometer and gas analyzer measurements by covariance maximization,
- Axis rotation for tilt correction using the planar fit method (Wilczak et al., 2001) with removal of velocity bias (Van Dijk et al., 2004), in running window mode (1-day window, updated every 30 minutes) (TEA sonic) and fixed period (7 days) mode (EC sonic),
- Flux quality check after Foken et al. (2004), selecting classes 0 and 1 for further analysis on a scale 0,1,2.

## 3 Results and Discussion

This section is organized as follows: (1) meteorological conditions during the experiment are presented, followed by (2) a characterization of mass flow control performance, a prerequisite for the (3) determination of concentration differences between accumulated updrafts and downdrafts, which, in combination with (4) vertical wind measurements, finally result in (5) trace gas fluxes. To inform the discussion on uncertainties of the eddy accumulation method, (6) coordinate rotation results, (7) uncertainties of vertical wind distributions, and (8) instrumental errors of the sonic anemometers and infrared gas analyzers used are presented.



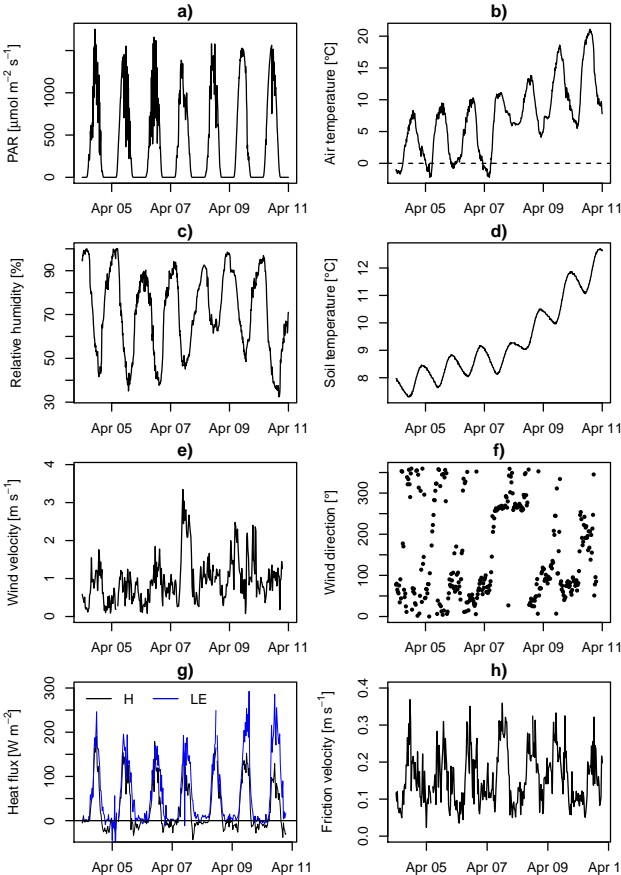

**Figure 6.** Meteorological conditions and turbulent energy fluxes during the experimental period from April 04 to April 10, 2015: Photosynthetically active radiation (PAR), air temperature, relative humidity, soil temperature, wind velocity, wind direction, sensible heat flux (H), latent heat flux (LE), and friction velocity. The wind and eddy covariance flux observations shown in Subfigures e)–h) were obtained from the R3 sonic of the eddy accumulation system in combination with the LI-7500 gas analyzer.

## 3.1 Meteorological conditions

Meteorological conditions (Fig. 6) during the experimental period from April 4 to April 10, 2015, were characterized by fair weather conditions with photosynthetically active radiation peaking at around $1500\,\mu\mathrm{mol\,m^{-2}\,s^{-1}}$ at noon (a). Air temperature was initially below 10 °C on the first day with frost during the nights but then rapidly increased to more than 20 °C on the last day (b). Similarly, there was a positive soil temperature trend (d). No precipitation was observed during this period. Wind direction (f) was dominated by easterly winds except for April 7 and 8 with mostly westerly and stronger winds (e) and higher relative humidity (c). April 7 stands out as the day with highest wind speed (e), high friction velocity (h), low radiation (a) and low sensible heat flux (g). The sensible and latent heat fluxes (g) otherwise largely tracked radiation levels with the highest latent heat fluxes observed on April 9 and 10 and with a decreasing Bowen ratio.



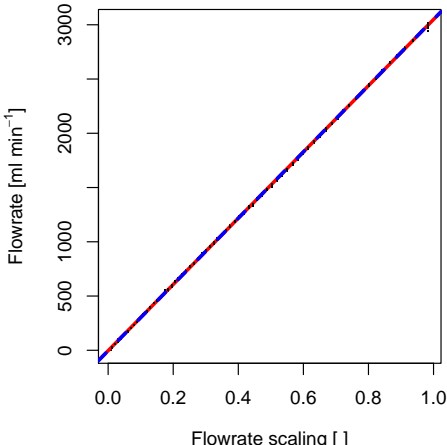

**Figure 7.** Linearity of two digital mass flow controllers verified by a conventional thermal mass flow controller for a series of 100 constant flow rate levels in the range of 0 to 3000 sml min$^{-1}$. The thermal mass flow controller reading is shown on the y-axis versus the set point of the digital mass flow controllers on the x-axis (black dots). The set point range on the x-axis from 0 to 1.0 is linearly related to a flow rate range of 0 to 3000 sml min$^{-1}$. Observed linearity errors of the digital mass flow controllers shown here are below 0.3% of the reading. Linear model fit of the digital mass flow controller of the "updraft channel" of the TEA system (red dashed line), and of the "downdraft channel" (blue dashed line), respectively.

## 3.2 Mass flow controller performance

Laboratory tests of the new digital mass flow controller used in this study showed that the accuracy and linearity of the digital mass flow controller for stationary flow was excellent: the maximum deviation of the new design from the conventional thermal mass flow controller used as reference was 0.3% over the full operating range (Fig. 7). The minimum and maximum flow rate of

5 the digital mass flow controller of 0.025 sl min$^{-1}$ and 3 sl min$^{-1}$ correspond to a dynamic range of 120, exceeding minimum performance requirements formulated by Hicks and McMillen (1984) for eddy accumulation, i.e. a dynamic range of 100 or higher. The tests also showed that the two controllers used for the updraft and downdraft channels of the TEA system, respectively, performed the same (red and blue line in Fig. 7).

We further compared the performance of the two mass flow controller designs under dynamic flow rate conditions (Fig. 8).

The new digital mass flow controller closely tracked the highly dynamic reference signal with tight error margins (uncertainty of digital mass flow controller shown in red around the black reference in Fig. 8), while the conventional thermal mass flow controller (blue line in Fig. 8) was unable to follow the dynamic reference signal, showing strong overshoot and overestimation of the reference flow rate.

Statistics of the mass flow controller comparison for the same data, as in the previous figure, are shown in the regression

plots in (Fig. 9). The new digital mass flow controller showed a perfect slope of 1.00, a high coefficient of determination of R$^2$=0.999 and a root mean square error (RMSE) of 8.8 sml min$^{-1}$ (Fig. 9 b), which is 50 times smaller than the RMSE of the




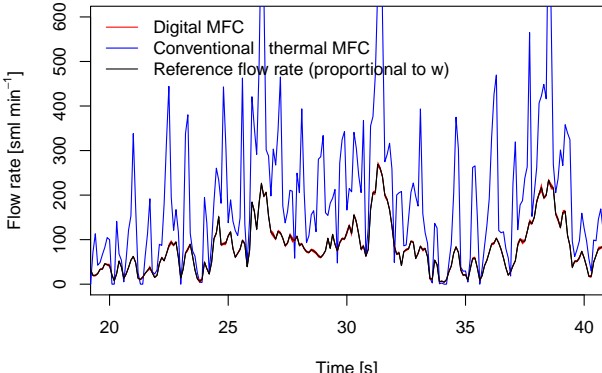

**Figure 8.** Mass flow rate readings of a conventional thermal mass flow controller (Voegtlin ready-smart series), in blue, measuring a dynamically changing reference flow (black). The magnitude of the reference flow rate, which is proportional to the magnitude of recorded vertical wind velocity data, varied at a frequency of 10 Hz. The reference flow (black) was generated using the new digital mass flow controller. The uncertainty range of the digital mass flow controller (red) is small. On the contrary, the mass flow reading of the conventional thermal mass flow controller deviates significantly from the reference as the conventional mass flow controller is unable to follow the highly dynamic reference signal.

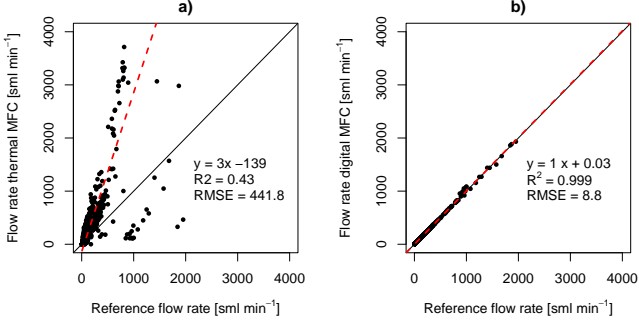

**Figure 9.** Regression of (a) the thermal mass flow controller flow rate versus the reference flow rate, and (b) the digital mass flow controller flow rate with errors versus the reference flow rate. The reference flow rate was varied at a frequency of 10 Hz proportional to measured vertical wind velocity data. 1:1 line (black solid line), linear model fit (red dashed line). Regression statistics: linear model equation; coefficient of determination, $R^2$; and root mean square error, RMSE.

conventional mass flow controller of $441.8 \, \mathrm{sml \, min}^{-1}$ (Fig. 9 a). The conventional mass flow controller further overestimated the reference by 300% and matched less than half of the variance of the reference signal ($R^2$=0.43).

In conclusion, the dynamic response, precision and accuracy of the digital mass flow controller are suitable for eddy accumulation sampling, while the conventional thermal mass flow controller is not.





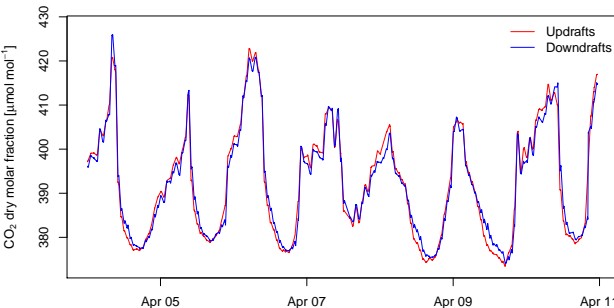

**Figure 10.** Dry molar fraction of $CO_2$ in the updraft and downdraft reservoirs of the true eddy accumulation device, in red and blue respectively. The difference in molar fraction is a result of the vertical $CO_2$ flux.

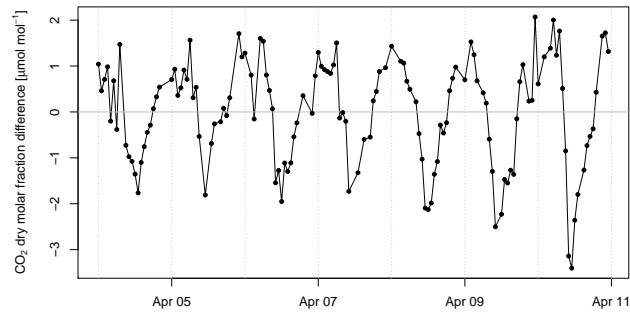

**Figure 11.** Difference in dry mole fractions of $CO_2$ between the updraft and downdraft reservoirs of the true eddy accumulation device. The '30-min raw' time series shown is the result of subtracting the mole fraction in the downdraft reservoir from the mole fraction in the updraft reservoir (Fig. 10). A positive $CO_2$ mole fraction difference indicates a $CO_2$ flux away from the surface (respiration) and a negative $CO_2$ mole fraction difference indicates a $CO_2$ flux towards the surface (assimilation).

### 3.3  $CO_2$ molar fraction and differences between accumulated updrafts and downdrafts

Time series of the $CO_2$ mole fraction obtained by the TEA system separately for the accumulated updrafts and downdrafts are shown in Fig. 10. Both the accumulated updrafts and downdrafts followed the common diurnal pattern of $CO_2$ mole fraction with minimal $CO_2$ densities during the day when photosynthetic activity of the vegetation is at its maximum, a gradual build up of $CO_2$ from the late afternoon through the night and finally a rapid decrease of $CO_2$ in the morning when the daytime turbulence removes nightly accumulation of trace gases and photosynthesis then further draws down the ambient $CO_2$ mole fraction. As expected, despite the generally similar course of the $CO_2$ mole fraction of the updraft and downdraft reservoirs, there was a small but systematic difference between the two with the $CO_2$ mole fraction of the updrafts (red line in Fig. 10) being lower than the downdrafts (blue line in Fig. 10). This difference was caused by the relative $CO_2$ depletion of updraft air due to photosynthesis during the day. At night, the inverse pattern was observed where updraft air was systematically enriched in $CO_2$ through respiration from soil and vegetation.





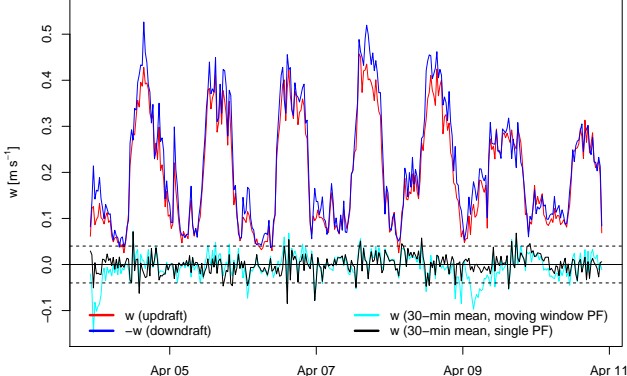

**Figure 12.** Vertical wind velocity of updrafts and downdrafts, in red and blue, respectively, averaged to 30-min resolution and shown as absolute values. Vertical velocity is subject to a 1 day running window real-time planar fit coordinate rotation as obtained from the TEA system. Mean vertical wind velocity at 30-min resolution after running window real-time planar fit coordinate rotation, in blue, and after a post-processing planar fit coordinate rotation with the fit period corresponding to the full period shown, in black.

The difference in $CO_2$ mole fraction between updraft and downdraft reservoirs is shown in Fig. 11. This difference was positive during the night and negative during the day. Windy conditions during the day cause a smaller magnitude of $CO_2$ difference as seen on April 7 (see Fig. 6 for wind and Fig. 11 for $CO_2$). Likewise, calm conditions enhance the $CO_2$ difference between updraft and downdraft reservoirs (see April 9 and 10, 2015, Fig. 11).

## 3.4 Mean absolute vertical wind velocity

Vertical wind velocity measurements from the TEA system are shown in Fig. 12, separately for updrafts and downdrafts (red and blue lines, respectively). Both updrafts and downdrafts show similar magnitude which is to be expected for a mean vertical wind velocity close to zero (black and cyan line in Fig. 12). On April 9 and 10, absolute vertical wind velocity $w$ during the day was lower than for other days (Fig. 12). Lower absolute $w$, indicating less vertical mixing, corresponded with more pronounced

differences in $CO_2$ molar fraction between updrafts and downdrafts, i.e. a more negative difference (Fig. 11) on the same two days. Under conditions of low winds and low turbulence, but intense radiation, the air close to the surface and the vegetation, which would be sensed as updrafts, was depleted in $CO_2$ through photosynthesis, relative to the air above.

Mean vertical wind velocity, $\overline{w}$, which ideally is zero over the 30-min flux integration intervals, rarely exceeded $\pm 4 \, \mathrm{cm \, s^{-1}}$ (black dashed lines), a threshold which, according to a simulation by Hicks and McMillen (1984), should not be exceeded to

avoid significant flux errors. On two occasions, $\overline{w}$ from the running window planar fit showed larger deviations from zero on April 4 and 9, 2015. Overall, the amplitude of $\overline{w}$ was smaller for the planar fit rotation using a single 7-day window compared to the running planar fit using 1 day windows.



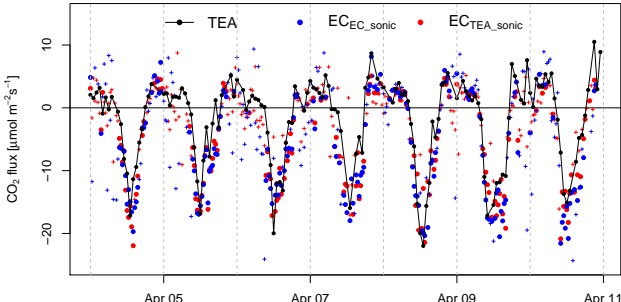

**Figure 13.** Comparison of $CO_2$ fluxes measured by true eddy accumulation (TEA) and eddy covariance (EC). Flux integration interval for both methods is 30 min. TEA fluxes were obtained every 60 min and EC fluxes every 30 min. The EC fluxes are quality flag filtered, accepting flags $\leq 1$ on a scale of 0 to 2 (Foken et al., 2004). The two alternative EC flux calculations shown, in blue and red, respectively, are from two separate Gill-R3 sonic anemometers using $CO_2$ density data from the same LI-7500 gas analyzer. The symbols indicate if EC flux estimates from the two sonic anemometers were within 50% from each other (full circles) or not (crosses).

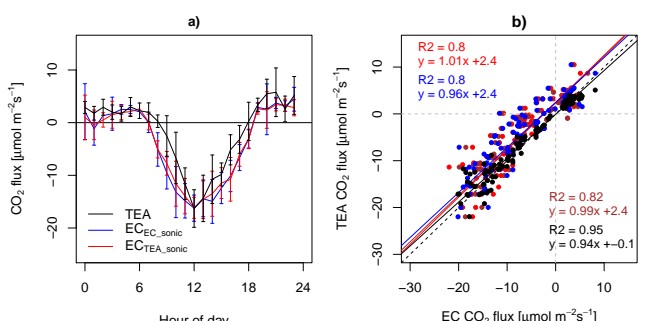

**Figure 14.** Mean diurnal cycle, a), and regression, b), of $CO_2$ fluxes measured by true eddy accumulation (TEA) and eddy covariance (EC) for the period 4–7 April, 2015. Flux integration interval for both methods is 30 min. TEA fluxes were obtained every 60 min and EC fluxes every 30 min. The two alternative EC flux calculations shown, in blue and red, respectively, are from two separate Gill-R3 sonic anemometers using $CO_2$ density data from a single LI-7500 gas analyzer. The EC fluxes are quality flag filtered, accepting flags $\leq 1$ on a scale of 0 to 2 (Foken et al., 2004) and filtered for consistency of EC fluxes from the two anemometers, accepting EC fluxes with a maximum relative flux difference of 50%. Error bars indicate $\pm 1$ standard deviation. b) Scatter-plot and linear model fit for TEA fluxes versus EC fluxes. The three EC flux versions shown are fluxes from the TEA sonic, from the EC sonic, and the mean of the fluxes from the TEA and the EC sonic, in red, and blue, and brown, respectively. Also shown are EC fluxes from EC sonic versus EC fluxes from TEA sonic, in black. Regression statistics shown are the coefficient of determination, $R^2$, and the linear model equation from a standard major axis regression (Legendre and Legendre, 1998; Sokal and Rohlf, 1995). The dashed black line is the 1:1 line.

## 3.5 $CO_2$ fluxes

The turbulent exchange of $CO_2$ between the vegetation and the atmosphere as observed by TEA is displayed in Fig. 13. Fluxes clearly show $CO_2$ uptake during the day (photosynthesis, with values up to ca. -20 $\mu mol\,m^{-2}\,s^{-1}$) and $CO_2$ release during the





night (respiration, with values up to ca. +5 $\mu mol\,m^{-2}\,s^{-1}$), see also mean diurnal cycle in Fig. 14 a). Temporal variability of $CO_2$ fluxes measured by TEA (Fig. 13) reflects variation in photosynthetically active radiation, PAR, (Fig. 6): April 8, 9, and 10, 2015 with high levels of radiation and the least amount of clouds also show the highest fluxes. April 6, which experienced more clouds during the day and therefore less abundant photosynthetically active radiation also showed relatively small $CO_2$

fluxes during the day. Similarly, on April 5 and 7, which were affected by clouds and reduced radiation in the early afternoon, the $CO_2$ fluxes during the afternoon were reduced compared to cloud free days, e.g. April 9.

Night time $CO_2$ fluxes measured by TEA showed a trend of increasing fluxes over the experimental period (Fig. 13). This trend corresponded to the observed trends of increasing air temperature and soil temperature (Fig. 6) over the same period. The observation of a positive correlation of positive $CO_2$ fluxes (respiration) with soil and air temperature is in line with the widely

accepted mechanistic understanding that soil respiration is a function of soil temperature with a positive correlation over the temperature range presented. Overall, from an ecophysiological point of view, the observed $CO_2$ fluxes corresponded well with their meteorological drivers.

From a methodological point of view, a key step in assessing the TEA method's performance is the comparison of the TEA method with the established EC method. Fig. 13 shows $CO_2$ fluxes measured by both TEA (black line) and EC (red and blue

points). The eddy covariance fluxes shown in Fig. 13 are quality filtered accepting flags $\leq 1$ (Foken et al., 2004, scale 0–2). This filtering reduced the number of available 30-min eddy covariance flux estimates to 90% and 93%, and filtering for flag $= 0$ reduced the fraction of available data to 45% and 46% for the sonic anemometers of the TEA and the EC system, respectively (both used for eddy covariance fluxes).

Generally, good agreement was observed between the TEA and EC methods. The differences between the TEA and EC

methods were of the same order of magnitude as the differences between the two EC flux estimates, which use the same infrared gas analyzer.

The intercomparison of the methods was complicated by the presence of high noise levels in the 30-min EC flux estimates. Analysis of R3 sonic anemometer raw data revealed that the EC sonic anemometer (blue line Fig. 13) was affected by correlated noise in the high-frequency wind and sonic temperature measurements, resulting in erroneously high sensible heat flux and

momentum flux estimates. The erroneously high variance of the horizontal and vertical wind components was particularly pronounced during the nights and decreased over the experimental period. On the contrary, erroneously high sonic temperature variance increased over the experimental period. $CO_2$ fluxes were affected to a lesser degree because the noise in the R3 sonic anemometer data was not necessarily correlated with the LI-7500 infrared gas analyzer measurements.

The sonic anemometer of the TEA system seemed less affected by noise. Therefore we used it as an alternative input to

the eddy covariance flux computations. However, the horizontal separation between the latter sonic anemometer and the open-path gas analyzer was large (0.7 m) relative to the low measurement height. Eddy covariance $CO_2$ fluxes from the TEA sonic anemometer were on average 18% smaller compared to the EC sonic anemometer. However, after eliminating inconsistent fluxes where the fluxes from the two sonic anemometers disagreed by more than 50% or alternatively 25%, eddy covariance $CO_2$ fluxes from the TEA sonic anemometer were on average 6% smaller and 2% larger, respectively, compared to the EC

sonic anemometer (Fig. 14).



A regression of $CO_2$ fluxes from the TEA method versus the EC method is shown in Fig 14 b), comparing TEA fluxes to the two alternative EC flux estimates after quality filtering eddy covariance fluxes with flags $\leq 1$ (Foken et al., 2004). Before applying the above consistency filter, the coefficients of correlation were $R^2$=76% and 67% and the linear model slopes were 1.04 and 0.87 when quality filtering eddy covariance fluxes for flags $\leq 1$ (Foken et al., 2004) and improved to $R^2$=71% and
76% and slopes of 0.99 and 0.92 for flags = 0, for TEA fluxes versus EC fluxes using the TEA and EC sonic anemometers, respectively. The correlation further improved when rejecting conflicting EC fluxes: limiting the difference of the two EC flux estimates from the two independent sonic anemometers to 50% of the mean of the two fluxes (flags $\leq 1$), the coefficients of regression of TEA versus EC fluxes increased further to 80%, 80%, and 82% with linear model slopes of 1.01, 0.96, and 0.99, for TEA fluxes versus EC fluxes using the TEA sonic anemometers, EC sonic anemometers, and their average, respectively
(case shown in Fig 14 b).

While a certain fraction of the observed deviations between TEA and EC flux estimates can be attributed to the methodological differences, the two independent eddy covariance flux estimates also showed deficiencies in agreement, with coefficients of determination of the EC to EC regression of $R^2$=84%, 86%, and 85% and slopes of 0.81, 0.82, 0.85, leaving 16%, 14% and 15% of the flux variance unexplained for quality flag filter thresholds of 2, 1, and 0. The percentage of additional unexplained
variance in the TEA fluxes with $R^2$=74% (regression of TEA fluxes versus mean EC fluxes from the two sonic anemometers, flag $\leq 1$) relative to EC fluxes with $R^2$=86% was 12% only. There was a positive intercept for the above regression cases between TEA and EC fluxes which ranged from 1.8 to 2.4 $\mu mol\,m^{-2}\,s^{-1}$.

We would expect from a side-by-side comparison of eddy covariance flux measurements using identical models of research class sonic anemometers and sharing the same gas analyzer, that the $R^2$ would exceed 90%. We interpret the compromised
match of the two current EC estimates largely as a result of compromised wind and sonic temperature measurements by the two R3 anemometers. When excluding the first four days which were relatively more affected by erroneous $w$, filtering for quality flags $\leq 1$ and again filtering EC fluxes to not exceed a relative difference between the two EC flux estimates of 50%, then the match of TEA versus EC $CO_2$ fluxes improved further, yielding $R^2$ values of 84%, 86%, and 86% and slopes of 1.04, 0.91, and 0.98 and intercepts of 2.0, 1.9, and 2.0 $\mu mol\,m^{-2}\,s^{-1}$ for TEA fluxes versus EC fluxes using the TEA sonic anemometers,
the EC sonic anemometers, and the mean of the TEA and EC sonic anemometers, respectively.

In relation to previous works on true eddy accumulation trace gas flux measurements, we note that, despite compromised data quality of one of the R3 sonic anemometers of the current study, the match of true eddy accumulation and eddy covariance $CO_2$ fluxes exceeded the match in any previously published true eddy accumulation experiments we are aware of. The closest results are those from Komori et al. (2004), who obtained a coefficient of determination for TEA versus EC $CO_2$ fluxes of
$R^2$=0.64, a slope of 0.95 and a relatively high intercept of 8.6 $\mu mol\,m^{-2}\,s^{-1}$ for 17 flux integration intervals.

## 3.6 Uncertainty of vertical wind measurements

Vertical wind measurements contribute to flux uncertainty in two ways: (i) instrumental errors of the sonic anemometer and (ii) non-ideal wind field and non-zero mean vertical wind velocity over the flux integration period.





### 3.6.1 Sonic anemometer measurement errors

Systematic and random errors in sonic anemometer measurements contribute to scalar flux uncertainty both in the EC and TEA methods. For a detailed analysis of sonic anemometer measurement errors we refer to instrument comparisons and quantifications of measurement errors for common sonic anemometer types (Loescher et al., 2005; Mauder and Zeeman, 2018; Foken

et al., 2019, and others). These studies suggest that measurement errors of sonic anemometers, including differences between different types of sonic anemometers, and differences between different units of the same type of sonic anemometer, may account for anywhere from several percent up to about 25% of the error in scalar flux measurements.

Regarding the R3-type sonic anemometer used in the current study, (Loescher et al., 2005, Fig. 5) found in wind tunnel tests, that the R3 sonic anemometer, like other post-mounted designs, suffered from flow distortion, systematically overestimating

vertical wind velocity. The R3 overestimated vertical wind velocity for vertical velocities below $0.15 \, \mathrm{m \, s^{-1}}$ by up to ca. 0.05 $\mathrm{m \, s^{-1}}$ for vertical velocities close to zero and underestimated vertical velocity by up to the same amount for vertical velocities up to $0.3 \, \mathrm{m \, s^{-1}}$ (gain error). In addition, when the stanchions supporting the upper transducers were in the flow path, the vertical wind velocity response was non-linear. Non-linearity and gain errors can result in misalignment of the coordinate system with the mean stream lines (see Sec. 3.6.2) and apparent asymmetry of vertical wind distributions (see Sec. 3.6.3).

### 3.6.2 Coordinate rotation

The following analyses present non-zero mean vertical wind velocities, which cause scalar flux uncertainty. Figure 15 shows vertical wind before and after coordinate rotation. Vertical wind velocity scaled by horizontal wind velocity ideally follows a (co)sine function when anemometer coordinates are tilted relative to stream line coordinates (Fig. 15 a). Planar fit rotation reduced the range of $\overline{w}$ from about $\pm 0.1 \, \mathrm{m \, s^{-1}}$ to about $\pm 0.05 \, \mathrm{m \, s^{-1}}$ (Fig. 15 b). The range of $\overline{w}$ was slightly smaller for the

7-day rotation window compared to the running 1 day window. Figure 15 c) indicates that before coordinate rotation the sonic coordinate frame was tilted relative to the stream lines, which followed the terrain slope. The slope of the relation of $\overline{w}$ over along-slope wind $U$ (c) vanished after planar fit rotation using the 7-day window (d). The running planar fit with 1 day window did not fully remove the dependence of $\overline{w}$ from along-slope wind velocity, biasing $\overline{w}$ by up to $\pm 0.02 \, \mathrm{m \, s^{-1}}$ over the range of along-slope velocities shown, corresponding to about 15% of bias before tilt-correction.

Fig. 16 presents residuals of mean vertical wind velocity $\overline{w}$ before and after coordinate rotation in $u$ and $v$ horizontal velocity space. This analysis relates $\overline{w}$ to wind direction, horizontal wind velocity and obstacles causing flow distortion. $\overline{w}$ in Fig. 16 a) before tilt correction shows the effect of the terrain slope on wind measurements in sonic coordinates. The slope effect was fully removed through planar fit rotation (Fig. 16 b). The 1-day running window planar fit led to a slight overcorrection of the tilt (Fig. 16 b) as already noted regarding (Fig. 15 d), which was not the case for the planar fit rotation using a 7-day window.

Residuals of $\overline{w}$ were small with the 1-day running planar fit, resulting in relatively larger $\overline{w}$ residuals up to about $\pm 0.05 \, \mathrm{m \, s^{-1}}$ (Fig. 16 b) compared to up to about $\pm 0.02 \, \mathrm{m \, s^{-1}}$ for the 7-day planar fit (Fig. 16 c).

Some dependence of $\overline{w}$ on horizontal wind velocity and direction was observed: for horizontal wind velocities of more than about $1 \, \mathrm{m \, s^{-1}}$, residuals of $\overline{w}$ were mostly positive, particularly for south-westerly and north-easterly winds (Fig. 16 b)





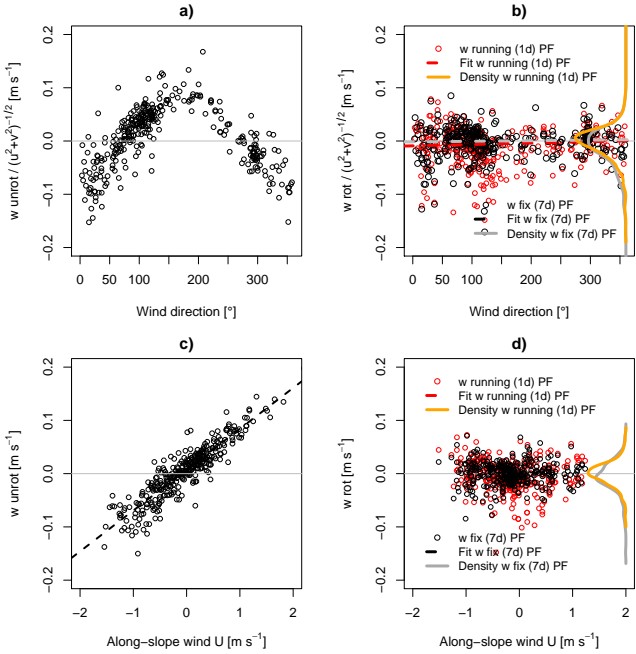

**Figure 15.** 30-min means of vertical wind velocity, $w$, before (a, c) and after (b, d) coordinate rotation as a function of wind direction (a, b), and as a function of along-slope wind, U, (c, d). The vertical wind velocity $w$ in a) and b) was is normalized by horizontal wind velocity. Subfigures b) and d) both differentiate between two approaches to the planar-fit coordinate rotation: the real-time planar fit applied in the TEA measurements deploying a moving window of one day ("running PF", in red and orange), and a single planar fit covering the full measurement period (seven days long) obtained in post-processing, ("fix PF", in black and gray).

and c). This was confirmed by the planar fit rotation of a 2-month long data set (Fig. 16 d). Possible interpretations include flow distortion from trees and bushes in the South-West and North-East and velocity dependent flow distortion of the sonic anemometer or nearby structures. No obvious influence of the stanchions of the anemometer on $\overline{w}$ was identified (Fig. 16 a–d).

### 3.6.3 Updraft-downdraft asymmetries

5    The TEA method requires $\overline{w} = 0$, which would result from symmetry in updraft and downdraft statistics. However, we observed asymmetry in the mean, the count and the sum of updraft and downdraft samples. Quantification of the flux uncertainty due to asymmetric distributions of $w$ would require co-spectral information of $w$ and $CO_2$ densities, which is generally absent for TEA measurements. Instead, we present a quantification of observed asymmetries of $w$, informing about the magnitude of the asymmetries and their variability over time.

10      Figure 17 shows that: (i) on average, the count of updrafts was larger than the count of downdrafts; however, (ii) on average, the mean of updrafts was smaller than the mean of downdrafts; and (iii) on average, the sum of updrafts was slightly smaller than the sum of downdrafts. It is noteworthy that the mean of $w$ can still be zero while the 7-day mean of the 30-min mean of





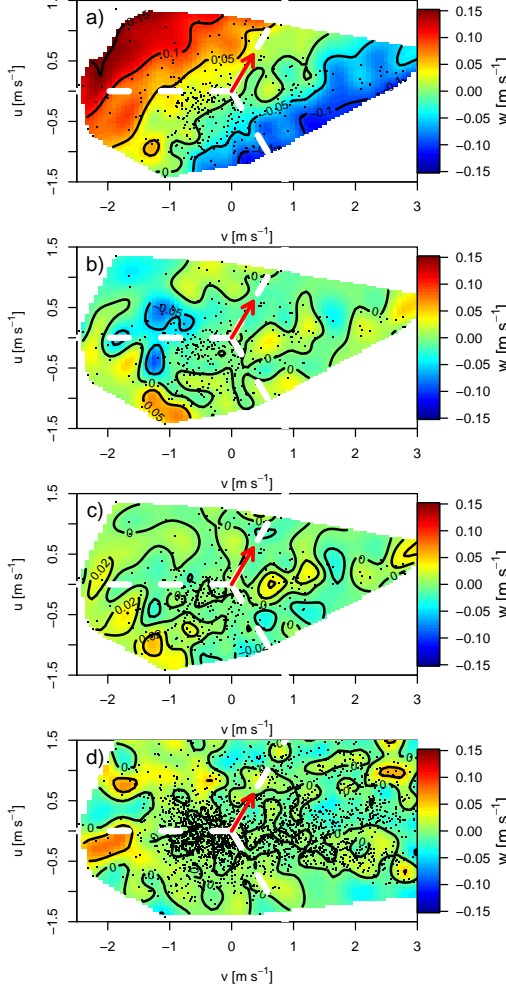

**Figure 16.** 30-min mean vertical wind velocity $w$ as a function of horizontal wind velocities $u$ and $v$ before and after planar fit coordinate rotation for two different planar fit procedures and two periods. Wind velocities were obtained by the Gill-R3 sonic anemometer which was used for true eddy accumulation. a) Unrotated vertical wind velocity; b) Rotated vertical wind velocity from real-time 1-day moving window planar fit rotation performed by TEA system; c) Rotated vertical wind velocity with single planar fit rotation period performed in post-processing; d) Same as c) but for longer period, i.e. from 1 April to 31 May, 2015. The first three Subfigures a) to c) show data from the experimental period of the current study from 4 April to 11 April, 2015. Black dots indicate location of individual 30-min mean vertical wind velocity readings in the u-v velocity space. Red arrows indicate the direction to North. Dashed lines, in white, indicate the azimuth of the vertical stanchions of the sonic head structure relative to the center of the sonic coordinate system.

updrafts and the 30-min mean of downdrafts is non-zero, or in other words, the ratio of the 30-min mean of the updrafts over the 30-min mean of the downdrafts is different from 1 as observed here. This can be understood by considering the different weights of updrafts and downdrafts in the mean akin to the asymmetry in the counts of updrafts and downdrafts, respectively.





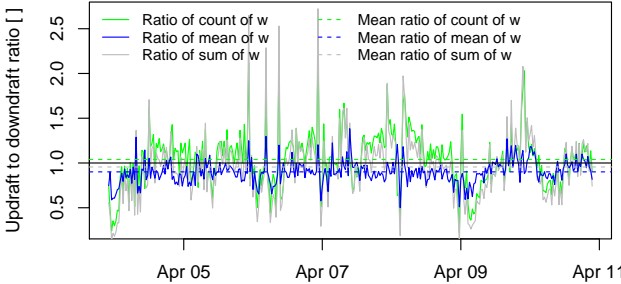

**Figure 17.** Ratio of statistical measures of vertical wind velocity of updrafts and downdrafts per 30-min flux integration interval. The statistical measures are the ratio of the count, the mean, and the sum of vertical wind velocity records during updrafts and downdrafts, respectively (solid lines). The dashed lines indicate the temporal mean of above statistics over the period displayed.

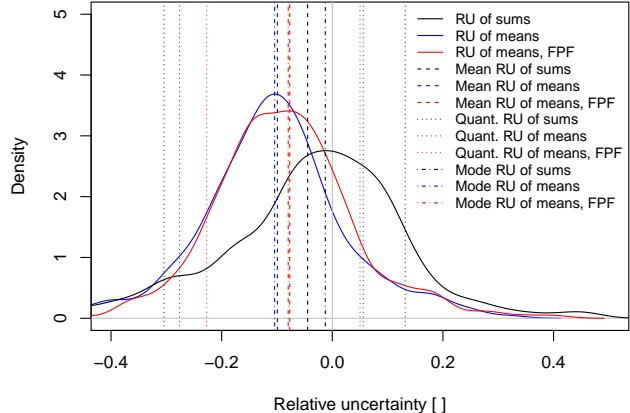

**Figure 18.** Probability density distributions of the relative uncertainty (RU) of the mean absolute vertical wind velocity, $\overline{|w|}$, due to non-zero mean vertical wind velocity over the 30-min flux integration periods (blue solid line). $\overline{|w|}$ is required for the determination of trace gas fluxes according to Eq. 2. The relative uncertainty of the mean vertical wind velocity was calculated as $RU = ((\overline{w_{up}} - \overline{|w_{down}|})/2)/((\overline{w_{up}} + \overline{|w_{down}|})/2)$, where the overline denotes the temporal mean of vertical wind velocity over the 30-min flux integration interval, and subscripts $up$ and $down$ refer to updrafts and downdrafts, respectively. Results based on $w$ from the real-time moving window planar fit coordinate rotation of the TEA system and from the fixed window post-processing planar fit rotation (FPF) are shown in blue and red, respectively. "Mean RU of means", "Mode RU of means", and "Quant. RU of means" indicate the mean, the mode, and the quantiles, respectively, for a probability of 10% and 90%, respectively. While mean vertical wind velocity $\overline{|w|}$ is needed for flux derivation, the sum of vertical wind velocity over the flux integration interval, $\sum |w|$, relates to the accumulated air sample volumes. The relative uncertainty of the sums, "RU of sums", corresponds to the contribution of vertical wind velocity to the volume mismatch correction defined in Sect. 2. Relative uncertainty of the sum of vertical wind velocity per 30-min flux integration interval was calculated as $RU = ((\sum w_{up} - \sum |w_{down}|)/2)/((\sum w_{up} + \sum |w_{down}|)/2)$. "Mean RU of sums", "Mode RU of sums", and "Quant. RU of sums" indicate the mean, the mode, and the quantiles of the distribution for 10% and 90% probability, respectively.





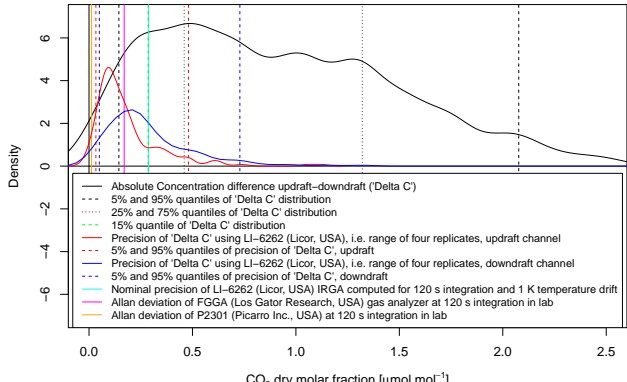

**Figure 19.** Density distributions and selected probabilities of the trace gas concentration difference signal and of the measurement uncertainties, and gas analyzer precision. The signal is the difference in molar fraction between accumulated updrafts and downdrafts (black line). The uncertainty of the measurements is expressed as the range of four replicated samples of the updraft and downdraft concentration, in red and blue, respectively. Vertical dashed lines indicate various probabilities of above distributions. Vertical solid lines indicate the nominal precision of the LI-6262 (LI-COR, USA) infrared gas analyzer used in this study, calculated for an integration time of 120 s (cyan). Precision of two types of laser spectrometers are also shown for reference: Allan-deviation of FGGA (Los Gatos Research, USA), in magenta, and G2301 (Picarro, USA), in orange, both determined in the laboratory.

Figure 18 shows probability density distributions of the relative uncertainty of the mean absolute vertical wind as defined in the figure caption. The distribution of the sums of updrafts and downdrafts is centered around zero with only a small negative bias of -0.01 for the mode of the distribution and a larger bias of -0.04 for the mean of the distribution. On the contrary, the distribution of the relative uncertainty of the means peaked for both the mode and the mean of the distribution at a more negative bias of -0.11, as observed for the running window planar fit (Fig. 18 blue line). Similar results were obtained for the 7-day stationary planar fit although with a smaller negative bias of 0.08 (Fig. 18 red line). Less than 10% percent of relative uncertainty values were more negative than -0.3 and less than 10% of relative uncertainties were larger than +0.14. In summary, updraft-downdraft asymmetries were on the order of 10% of the mean absolute vertical wind velocities used in flux calculations.

### 3.7 Uncertainty of trace gas concentration measurements

Trace gas flux errors are a function of accuracy and precision of the trace gas analysis. Regarding accuracy, bias between the two infrared gas analyzers used for the TEA and EC methods, respectively, is more relevant than absolute accuracy for comparing the TEA method to the EC reference method. By comparing time averaged time series of $CO_2$ concentrations of the LI-6262 and LI-7500 infrared gas analyzers, a time variable bias was found which accounted for up to 5% of the scalar flux.

Regarding scalar flux errors from the TEA method, systematic errors leading to bias between measurements of updrafts and downdrafts and precision are important. Systematic errors biasing the concentration difference between updrafts and downdrafts are difficult to quantify. The following results quantify the precision of the gas analysis, based on analysis of four





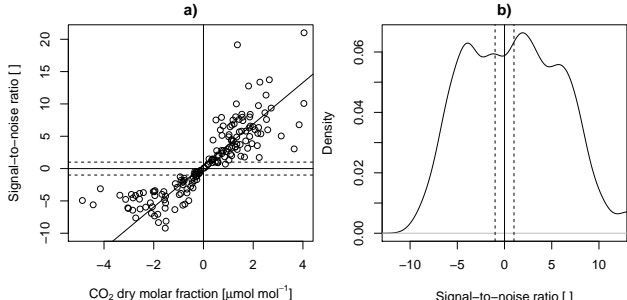

**Figure 20.** (a) Signal-to-noise ratio ($SNR$) of the trace gas dry molar fraction difference measurements as a function of the this difference, the latter being proportional to the trace gas flux. The solid line is a linear model fit using ordinary-least-squares regression. (b) Density distribution of the signal-to-noise ratio. Dashed lines in (a) and (b) indicate a signal-to-noise ratio of $\pm 1$, corresponding to probabilities of 37.7% and 49.5%, respectively. Consequently, in 88% of the cases the signal-to-noise ratio is higher than one, leaving 12% of the cases unresolved. The display in (a) excludes one extreme value at $SNR = -66.7$.

replicated measurements of 120 s each of the accumulated updraft and downdraft concentrations, per 30-min flux integration interval. The results comprise precision of the gas analyzer at the measurement frequency of 1 Hz as well as precision of the TEA gas sampling, storage and delivery system, feeding samples to the gas analyzer. The latter includes drift of the gas analyzer signal and of the trace gas concentration over the time required to determine a concentration difference between updrafts

and downdrafts, i.e. two times 150 s.

Regarding the observed $CO_2$ concentration signal, 90% of $CO_2$ dry molar fraction differences between updrafts and downdrafts at 30-min integration were between 0.14 and 2.08 µmol mol$^{-1}$ (Fig. 19). Regarding the observed precision of the total gas analysis system under field conditions, for 90% of flux integration intervals, the $CO_2$ dry molar fraction measurements over four replicated measurements of the updraft reservoir varied in the range of 0.033 to 0.48 µmol mol$^{-1}$. The precision of

downdraft measurements was 50% lower with 90% of the downdraft measurements showing a range of the four replicates of 0.05 and 0.73 µmol mol$^{-1}$.

For 85% of the flux integration intervals, the signal, i.e. the dry molar fraction difference between updrafts and downdrafts, was larger than the nominal and extrapolated 120-s precision of the LI-6262 infrared gas analyzer used in this study, as well as the precision, i.e. Allan deviation, of two laser spectrometers we tested in the laboratory. The latter two instruments were not

used in the current study but characteristics are provided to put the instrument used in this study into perspective to current state-of-the-art greenhouse gas monitors. Note that the indicated precision of the LI-6262 of 0.29 µmol mol$^{-1}$ is an extrapolation of nominal precision and drift values to 120 s, where nominal precision was given as peak-to-peak noise, rather than Allan deviation, which was used to characterize the laser spectrometers and is by definition smaller than or equal to the peak-to-peak noise.

Also note that the Allan deviation at 120 s integration of the G2301 (Picarro) instrument of 0.0125 µmol mol$^{-1}$ appeared to be only 7.4% of the Allan deviation of the FGGA (Los Gatos Research) laser spectrometer of 0.17 µmol mol$^{-1}$. However, in





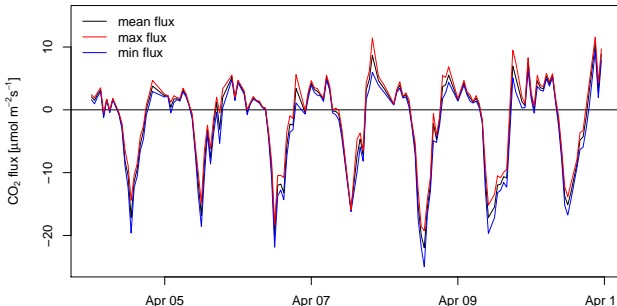

**Figure 21.** Range of $CO_2$ trace gas flux estimates observed by the true eddy accumulation method, accounting for the range of possible flux estimates from four replicated measurements of the dry molar fraction difference between accumulated updrafts and downdrafts per 30-min flux integration interval.

addition to differences in the design of the two spectrometers and any potential differences in test conditions, it appears from the analysis of Allan deviation that the G2301 (Picarro) instrument may be subject to some degree of internal smoothing of the gas concentration readings. We cannot say with certainty to which degree such potential smoothing might have affected the Allan deviation at 120 s integration time, because the manufacturer of the instrument was unable to provide further information

on the suspected filter beyond acknowledging its existence.

The observed signal-to-noise ratio ($SNR$) of the total trace gas analysis system under field conditions ranged between -9 and +21 (and one value at -66.7) and improved with the magnitude of the signal itself (Fig. 20 a). Slope and intercept of an ordinary-least-squares linear model fit to this relation were 3.2 $\mu$mol$^{-1}$ mol and 0.48, respectively. The fact that this quasilinear relationship and the slope significantly differed from zero means that larger fluxes have relatively smaller errors of the type

considered, a feature reducing absolute uncertainty of trace gas flux budgets.

For 88% of 30-min flux integration intervals, the signal-to-noise ratio was larger than one (Fig. 20 b). Over the period of the experiment, the sum of the same noise data as above accounted for up to 25% of the sum of the trace gas concentration signal, i.e. of the difference in $CO_2$ dry molar fraction between updraft and downdraft reservoirs. The 25% is a maximum estimate for this type of noise, as it was determined from the range of the four replicates of concentration difference measurements, which

is sensitive to extremes.

### 3.8 Uncertainty of trace gas flux measurements

The uncertainty of trace gas fluxes due to the uncertainty of the gas analysis is shown in Fig. 21. Over the period of the experiment, the sum of the noise range, i.e. the absolute value of the difference between the largest and smallest flux estimate, accounted for 37% of the sum of the signal, i.e. the absolute value of the mean flux. As stated above for $CO_2$ concentrations,

the 37% is a maximum estimate of this type of noise, because it was determined from the range of the four replicates of concentration difference measurements, which is sensitive to extremes, and because this estimate is additive. In practice, it is



highly unlikely that this uncertainty range leads to additive errors, instead, some of the errors would cancel, leading to much smaller actual uncertainties.

Regarding the time series of trace gas flux noise, i.e. the range of maximum and minimum flux estimates (Fig. 21), calm conditions with low wind speeds and low friction velocities, e.g. on April 10, result in relatively large concentration differences

and relatively small vertical wind terms contributing to the trace gas flux calculations and therefore result in relatively low uncertainty of the flux due to the uncertainty of the gas analysis. The opposite can be observed for windy conditions with high friction velocity, e.g. on April 7, which result in relatively small concentration differences and a relatively high contribution of the vertical wind term to the flux calculations and therefore a relatively high uncertainty of the flux due to the uncertainty of the gas analysis.

The total uncertainty of the trace gas flux needs to account for the uncertainty of the mass flow control, the uncertainty of the concentration differences and the uncertainty of the vertical wind signal. Ideally, such analysis would incorporate the effect of different approaches of coordinate rotation not just on the residuals of $\overline{w}$ but also the effect on the fluxes themselves. This would require consideration of co-spectral information of wind and scalar using time-resolved high-frequency data and simulations of true eddy accumulation with different coordinate rotation approaches.

**4 Conclusions**

The following conclusions intend to: summarize the performance of the true eddy accumulation method, put the results of the current experiment into context relative to existing published studies, summarize and quantify main sources of uncertainty, report on limitations and lessons learned during the current experiment, suggest future improvements regarding technical and methodological aspects, and finally to identify applications where true eddy accumulation can facilitate novel flux measure-

ments in the future.

The current study has presented $CO_2$ fluxes measured by true eddy accumulation. The TEA system measured continuously and automatically fluxes at 30-min resolution over a duration of more than seven days. The TEA measurements were able to capture fluxes representing the biological activity of the system. TEA flux measurements compared favorably with eddy covariance reference measurements with $R^2$ values of up to 86% and a regression slope of 0.98.

A novel implementation of dynamic mass flow control was key to the success. It was 50 times more accurate in terms of root mean square error than the conventional thermal mass flow controller reference during laboratory tests and proved to be robust and without failure during more than three years of operating time in the field. Further innovative features were the digital signal processing and the real-time sampling decisions incorporating on-line coordinate rotation and correction of the mean vertical wind and finally, the elimination of dead-volumes in the gas sampling system.

Compared to earlier studies published on true eddy accumulation flux measurements (Desjardins, 1977, on temperature fluxes, Speer et al., 1985, and Neumann et al., 1989, on water vapor fluxes, Rinne et al., 2000, on isoprene fluxes, and Komori et al., 2004, on $CO_2$ fluxes), the current study obtained the best fit of TEA fluxes to EC fluxes of any trace gas or scalar. The current study also presents the longest continuous $CO_2$ flux measurements by TEA.





A detailed analysis of uncertainties of the TEA method was presented in terms of the uncertainty of the mass flow controllers, the uncertainty of the trace gas handling and analysis system, and the uncertainty of the vertical wind velocity measurements and 30-min means. Uncertainties of the eddy covariance method and instruments were partially quantified through two replicated flux computations using two alternative sonic anemometers. Uncertainties of the EC fluxes explained a significant fraction

of the mismatch between the TEA and EC methods. The signal-to-noise ratio of the TEA trace gas analysis system allowed to detect the concentration difference signal in 88% of 30-min flux intervals. Maximum uncertainty estimates of the TEA trace gas measurement precision accounted for up to 25% of the concentration differences and up to 37% of the fluxes. A comparison of the precision of three gas analyzers suggests that deployment of state-of-the-art laser spectrometers would significantly reduce TEA flux uncertainty due to uncertainties in the gas analysis with preliminary analysis suggesting an improvement in

precision by a factor of 10 or more for some instrument models. This would likely reduce the flux uncertainty due to the gas analysis to about 5% or less. Residual mean vertical wind velocities were generally smaller than $0.05\,\mathrm{m\,s}{-1}$. Uncertainties of the mean of absolute vertical wind velocities, which are needed for flux calculations, in terms of undesired residuals of mean vertical wind velocities after coordinate rotation were frequently on the order of 5%. The uncertainties of the mass flow control were relatively small compared to uncertainties of the gas analysis, uncertainties of residual mean vertical wind velocities and

uncertainties of the eddy covariance flux estimates.

The following two design limitations were discovered: firstly, the continuous and long-term operation with frequent charging and discharging of the air sampling bags with on the order of 1500 charge cycles per month over time lead to increasing levels of fatigue of the material and in turn after a few weeks to a significant amount of leakage and therefore contamination of the samples with ambient air. The second observation relates to the intermittent nature of the gas flow, the variable accumulation

volumes, and the intermittent gas analysis in the current bag based accumulation design. Intermittent operation causes instationarity of the following parameters: air pressure in the gas handling system, temperatures of air and system components, and interactions of air constituents with the internal surfaces of the device such as adsorption and desorption of gas molecules at internal surfaces. Instationary conditions can lead to signal drift, and variation of moisture content and subsequently to less accurate flux measurements.

To address the above mentioned limitations, the author suggests to explore the idea of a new system design for TEA using rigid air containers of constant volume and with continuous-flow operation replacing flexible air bags. In such a new design the charging and discharging of the air reservoirs would happen continuously and at the same time. This new design principle would overcome the issue of material fatigue and compromised accuracy due to instationarities in the operation. A key methodological advantage of the new continuous-flow design is furthering the opportunity to merge the principles of true eddy accumulation

sampling with eddy covariance sampling simultaneously with the very same measurement device, the same air samples and the same gas analyzer.

Using a precise state-of-the-art laser spectrometer the author has since implemented such a continuous-flow system suggested above and demonstrated its superior performance compared to conventional discrete cyclic charging of air bags. True eddy accumulation $CO_2$ fluxes observed with the new continuous-flow system were tightly correlated with eddy covariance

fluxes with $R^2$ values of up to 96%. More details on the latter study will be reported separately.



The impact of coordinate rotation on true eddy accumulation fluxes has been discussed. We have suggested a new type of coordinate rotation, which we refer to as "surface fit". Similar to the planar fit method, it aligns the coordinate system with the mean stream lines, accounting for a multi-dimensional parameter set including wind direction, flow distortion and optionally other independent variables in an integrative, continuous way.

We would like to highlight the need for research on flux corrections for TEA in a comprehensive way similar to the body of work which exists on EC flux corrections. Future work needs to investigate and establish flux corrections specifically for the TEA method, including the equivalent to the correction of trace gas fluxes due to density fluctuations caused by simultaneous transfer of heat and water vapor (Webb et al., 1980). The derivation of this and other corrections specifically for the TEA method is non-trivial and will be addressed in separate work.

The current implementation of TEA suggests that this method has the potential to facilitate flux measurements of trace gases and other atmospheric constituents for which no fast gas analyzers are available. TEA is an alternative when the precision and accuracy of currently available analyzers is insufficient for high frequency EC applications. The low power consumption of the current TEA systems with low sample flow rates will enable new applications, including off-grid use in solar and battery powered stationary and mobile applications. The long sample integration times give TEA a further advantage over EC,
allowing for simpler analyzer design compared to high frequency analyzers at the same precision or alternatively providing ultimate precision through long integration times when using a high quality analyzer.

It is evident that $CO_2$ fluxes in particular can be readily observed with alternative methods. However, the non-reactive and non-polar trace gas $CO_2$ is an ideal candidate to assess the performance of the TEA method. The current experiment is a successful proof-of-concept demonstrating that true eddy accumulation with dynamic and accurate air sampling proportional
to vertical wind velocity can be achieved in practice today. The lessons learned during the present work provide concise avenues including above outlined machine design considerations and required flux corrections for further improving the true eddy accumulation method to enable accurate and reliable flux measurements of more trace gases and atmospheric constituents than ever before.

*Author contributions.* This study was conceptualized, designed, performed, analyzed, interpreted, managed, and written by L.S. with contri-
butions to post-processing of gas analyzer data, to TEA flux calculations, and to discussions on the methods by A.E. The mass flow control system, used in this study, was conceptualized, designed, built, and calibrated by L.S.

*Competing interests.* The authors declare no competing interests.

*Disclaimer.* Any statements about the performance of measurement devices presented in this study, including commercially available gas analyzers, sonic anemometers, or mass flow controllers, are based on observations from the actual units deployed in the current study and do
not claim general validity for other types or different units of the same type or under different conditions.



*Acknowledgements.* We gratefully acknowledge the support of the Bioclimatology group, Prof. A. Knohl, University of Goettingen, in particular technical assistance by M. Puhan, D. Fellert, F. Tiedemann and H. Kreilein during calibration, preparation and setup of instruments in the field and for the provision of meteorological measurements. We further acknowledge Dr. J. Braden-Behrens and Christian Markwitz for fruitful discussions during preparation of the manuscript. L.S. would like to thank Prof. Th. Foken for exposure to the theory of eddy

5    accumulation flux measurements as well as B. Burban, J.-Y. Goret, Dr. J. Morison, Dr. M. Perks, and D. Schreiber for their support and trust along the way.



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
