# Peer review of "True eddy accumulation trace gas flux measurements: proof-of-concept"

_Atmospheric Measurement Techniques, 2018_

## Referee Comment (RC1) · Anonymous Referee #2 · 15 Apr 2019

The paper needs several clarifications with respect to the experimental setup, the performed calculation of TEA fluxes and some results given in Fig.12-16.

Page 3, Line 10: The paper is now presented by 2 authors. L 23: The paper is now presented by 2 authors. Please give the reference to such studies.

P4: Fig.1: Please refer to section 1.1.3 Define DEA

L4: See page 3 Authors

P6, L15: Please explain the meaning of noise in this context ?

P6, L27-28: $\beta$ varies between max. 0.3 and about 0.8 in special conditions. Please explain "the order of magnitude lower accuracy" due to $\beta$ approach.

[Figure]

P6, end: "aggressive use" ? noise ?

P7, L11: noise in the flux ?

If somebody states a flux is uncertain, he must refer to the reference standard. What is the reference standard for a flux ?

P9, Items 1-3: Please add in the text or cite

P9-11: Please give a detailed description and diagram of the experimental setup of the TEA as it was used. For example, there is no information about i.e. the position of the flow controllers in the tubing, the pump, and no estimation of the influence i.e. of delays, stages of pressure drops and dead volumes. (Also section 3.2)

P14, L4: Please add the unpublished work in an appendix.

At the end of this section, please present exactly the method applied in the presented study.

Section 2.7: You have not shown any data about flow distortion for R3 and did not correct for any flow distortion. What is your estimated offset of R3 in w - axis ?

P16, Eq.7,8: if mean w = 0 the $F_c = 0$ ?

The TEA relies on : $\sum w^+ c^+ - \sum w^- c^- = \overline{w'c'}$.

Here it is replaced by eq.7 . Where is the difference to REA ?

In Fig.12 which w is used for TEA flux calculations?

In Fig.15 and 16 you show that the wind vector seems to follow approximately the terrain. This is a nice example for along slope wind and should be also discussed with respect to influence of the rotation method.
* * *

---

## Referee Comment (RC2) · Anonymous Referee #1 · 29 Apr 2019

Review of the manuscript amt-2018-460 "True eddy accumulation trace gas flux measurements: proof-of-concept" by Lukas Siebicke and Anas Emad

The authors present a new setup for performing True Eddy Accumulation (TEA), where air is sampled separately for updrafts and downdrafts at a flow proportionally to the magnitude of vertical wind speed. This allows for measuring fluxes of constituents where no fast sensors are available that allow direct eddy covariance measurements. Other than Relaxed Eddy Accumulation (REA), TEA is a direct flux method and has therefore less theoretical limitations (e.g. scalar similarity, turbulent characteristics). The main technical advancement that allowed performing the TEA measurements was the development of a mass flow controller, that is capable of regulating the flow at 10 Hz to resolve the turbulent motion. Unfortunately, this work just gives examples of the performance of this new device, but no detailed technical descriptions or a more detailed description of the working principle of the mass flow controllers used. For the advancement of the TEA-technique, it is important that this information will be available in an appropriate way, soon.

Nevertheless, the authors are able to show the superior performance of their TEA-setup with respect to prior attempts in comparison to eddy covariance measurements and perform a thorough error analysis. Furthermore, they discuss in depth the major error sources and give an outlook on how to further improve the performance of the method. The manuscript is generally well written, well structured and the results are illustrated by a sufficient number of figures. Therefore, I support publication after considering the minor revisions given below.

General comments:

The manuscript was originally submitted by one author. Formulations need to be adapted (e.g. I => we).

The authors showed that the flow controllers could reproduce the correct flow with respect to a reference thermal mass flow controller and that they can resolve 10 Hz, but what about a zero offset (leak)? I think it is important to show here that there is really zero airflow if the other channel is sampled.

Specific comments:

P3, L26: $R^2$ is the coefficient of determination

P8, point 5: As an idea for improvement:

Would it be possible to place a fast thermocouple into the sample airstream and correlate this to the sonic temperature? It might not work for real time correction of the sampling, but give an estimate on the magnitude of decorrelation for post processing.

P9, point 1: As the working principle of the mass flow controller has not been mentioned it is difficult to judge if there are issues in the mass flow control due to the effect of water vapor (e.g. (Lee, 2000). If there is a latent heat flux moisture in the updrafts and downdrafts must be different. Can the authors comment on how this could affect the volume mismatch?

P10, L5: Please provide coordinates.

P14, L16-18: Do not fully understand this sentence please consider revising.

P16, eq. 7: If $\bar{w} = 0$ => Fc =0. Therefore, it must be either $\left|\overline{w^+}\right|$ or $\left|\overline{w^-}\right|$ (as they are of equal magnitude. Please clarify.

P17, eq. 10: Is $\bar{w}$ here the mean vertical velocity (for the averaging period) or is it the "mean of the absolute value of vertical wind velocity". Please clarify.

P18, first paragraph: Would be very valuable to have a schematic sketch of the setup explaining length of inlet lines, position of flow controllers and sampling bags including dead-volumes.

P20, L14: Which type of embedded computer?

P25, L10: How was the reference signal generated? => Description was hidden in the figure caption of Fig. 8. Please include in text as well.

P29: Please enlarge figures 13 and 14. In Fig. 13 the symbols, weather crosses or circles as described in the legend, are not identifiable. In Fig. 14 it is difficult to distinguish the different lines.

Furthermore, from Fig. 14 a) it seems that TEA is systematically underestimating the fluxes during day, which is not so clear looking at panel b and especially the slopes of the linear regressions. Although I understand the reasoning for using standard major axis regression for two independent variables it would be interesting to see how a "standard" linear regression would look like. I think it can be justified to use the EC flux as the "controlled variable" by the statement that EC serves as a reference method.

P30, L19-21: From Fig. 14 it seems that this statement needs to be clarified. The two EC-flux estimates agree much better than the EC and the TEA measurements.

P31, first paragraph: It would be good to name all cases always in the same order or mark them with indices.

P33, Fig. 15: In Panel b the legend is hardly distinguishable from the data-points.

P34, last line: "akin"? Not clear what this word means in this context. Consider revising.

P36, L6:"negative bias" => - 0.08

References:

Lee, X.: Water vapor density effect on measurements of trace gas mixing ratio and flux with a massflow controller, J. Geophys. Res. Atmos., doi:10.1029/2000JD900210, 2000.

---

## Author Comment (AC1) · 11 Jun 2019

**The paper needs several clarifications with respect to the experimental setup, the performed calculation of TEA fluxes and some results given in Fig.12-16.**

**Page 3, Line 10: The paper is now presented by 2 authors.**

**AR:** We have revised the text here and in other places throughout the manuscript to reflect the update of the list of authors (now two authors) as also noted by the second referee.

**L 23: The paper is now presented by 2 authors. Please give the reference to such studies.**

**AR:** Regarding the authors, the text has been revised (see also previous comment).
Regarding the references to „such studies": Mentioned studies have currently been published as conference contributions only but will be published later as peer-reviewed articles. In the current version of the manuscript commented by the referee, asking for references, we followed the explicit instructions given by the handling Editor, who asked us to remove any references to conference contributions (as can be seen in the online review process). Therefore we are unable to give further references to such studies. Instead we have updated the text to reflect this situation and mention the respective works as 'unpublished'.

**P4: Fig.1: Please refer to section 1.1.3 Define DEA**

**AR:** We have added to Fig.1 a reference to section 1.1.3 defining DEA as suggested.

**L4: See page 3 Authors**

**AR:** We have revised the text accordingly, see also comment above (Page 3, Line 10).

**P6, L15: Please explain the meaning of noise in this context ?**

**AR:** We clarified the meaning of noise in the revised manuscript. The revised text expresses the impact of scalar similarity and the dead-band on flux errors as simulated by Ruppert (2002).

**P6, L27-28: β varies between max. 0.3 and about 0.8 in special conditions. Please explain "the order of magnitude lower accuracy" due to β approach.**

**AR:** "The order of magnitude lower accuracy" was cited from the reference given (Foken 2008, on page 135 of the previous edition Foken 2003) and reference therein (Ruppert et al., 2002). The order of magnitude does not mean that the flux is uncertain due to the range of beta β but rather to the choice of β as either a constant or as a variable obtained from a proxy simulation. For the current manuscript the essential information is not the specific indication how much more uncertain the flux from a constant β would be but rather just the fact that a variable β is more appropriate as it reduces flux uncertainty. We have revised the text of the manuscript to express this.

**P6, end: "aggressive use" ? noise ?**

**AR:** The text of the revised manuscript has been rephrased and is now more specific.

**P7, L11: noise in the flux ?**
**If somebody states a flux is uncertain, he must refer to the reference standard. What is the reference standard for**

**a flux ?**

**AR:** The reference for the uncertainty of disjunctly sampled signals is the continously sampled signal (in this case of both the scalar and the vertical wind velocity, from which the fluxes are derived). We have revised the text of the manuscript to express this. Thanks for the indication that the reference should be stated explicitly.

**P9,Items 1-3: Please add in the text or cite**

**AR:** Regarding 1, we have added a reference to the relevant manuscript section, regarding 2 we have expressed the current status of the work and a reference to the literature, and regarding 3, we have added in the text a proposed correction approach.

**P9-11: Please give a detailed description and diagram of the experimental setup of the TEA as it was used. For example, there is no information about i.e. the position of the flow controllers in the tubing, the pump, and no estimation of the influence i.e. of delays, stages of pressure drops and dead volumes. (Also section 3.2)**

**AR:** We have followed the suggestion of the referee and included a detailed technical description of the system in an additional figure, i.e. Fig. 5 of the revised manuscript. The diagram of the experimental setup also includes the position of the flow controllers in the system, the tubing, the pumps, the delays, stages of pressure drops and dead volumes. The revised text also contains further details on the positioning of the air inlets relative to the sonic anemometer.

**P14, L4: Please add the unpublished work in an appendix.**
**At the end of this section, please present exactly the method applied in the presented study.**

**AR:** We have focused the text in the revised manuscript: now the principal idea of fitting a surface is mentioned directly insitu in the text and a reference to a related publication by Ross (2005) is given. An appendix presenting a new coordinate rotation method would be out of the scope of the current manuscript and in any case not appropriate as the manuscript does not use this method. The exact method used in the current study is presented in section 2.8.3 in the third paragraph, including citations.

**Section 2.7: You have not shown any data about flow distortion for R3 and did not correct for any flow distortion. What is your estimated offset of R3 in w - axis ?**

**AR:** Given the current data set it would be difficult to partition w residuals into (i) flow distortion, (ii) zero velocity offset, and (iii) non-zero mean vertical wind velocity during a given observation interval (such as the planar fit interval of one day). Therefore we estimate offset of the R3 related specifically to flow distortion by refering to a study by Loescher (2005), who found in a wind tunnel experiment that the vertical velocity bias of the R3 was 0.04 m s-1 at 0 m s-1 vertical wind velocity and -0.05 m s-1 at 0.3 m s-1 vertical wind velocity relative to a hotfilm reference.

P16, Eq.7,8: if mean w = 0 the $F_c = 0$ ?
The TEA relies on : $\sum w^+ c^+ - \sum w^- c^- = \overline{w'c'}$.
Here it is replaced by eq.7 . Where is the difference to REA ?

**AR:** Please note that Eq. 7 of the original manuscript, which you reference, indicates to first take the absolute value of *w* and then apply the temporal mean, which is denoted by the overbar, which includes the absolute value! This means the order of the mathematical operations is important! We believe that the manuscript is already correct. To avoid further misinterpretation, we have included a sentence to explicitly alert the reader.
Regarding the difference of Eq. 7 to REA, we have clarified this in the revised manuscript through addition of the corresponding REA formula (see Eq. 10 of the revised manuscript).

**In Fig.12 which w is used for TEA flux calculations?**

**AR:** Note that the fluxes were calculated from $\overline{|w|}$ according to Eq. 7, i.e., not using updrafts or downdrafts separately, as shown in Fig. 12 in red and blue. We have added this note to the caption of Fig. 12 to be clear.

**In Fig.15 and 16 you show that the wind vector seems to follow approximately the terrain. This is a nice example for along slope wind and should be also discussed with respect to influence of the rotation method.**

**AR:** The influence of the rotation method on the along slope wind vector is discussed here: P32 L 22-24 in the original manuscript. In the revised manuscript we have added a further subfigure reference to the text. We also noticed that by mistake the regression lines were missing from Fig.14 Subfig. d). We have added the regression lines to the figure in the revised manuscript. The figure now clearly shows that the 7-day planar fit results in a vertical velocity independent of along-slope wind U, whereas the 1-day planar fit does not, biasing 30-min mean w by up to +- 0.02 ms-1. This addresses the referee's question on the influence of the rotation method.

References:

Foken, T.: Angewandte Meteorologie, Springer, pp. 298, 2003.

Foken, T. and Napo, C. J.: Micrometeorology, vol. 2, Springer, pp. 362, 2008.

Loescher, H. W., et al. Comparison of temperature and wind statistics in contrasting environments among different sonic anemometer–thermometers. *Agricultural and forest meteorology*, 2005, 133. Jg., Nr. 1-4, S. 119-139.

Ruppert, J, Wichura B, Delany AC, Foken T (2002) Eddy sampling methods, A comparison using simulation results, 15th Symp on Boundary Layer and Turbulence, Wageningen, 15-19 July 2002, Am. Meteorol. Soc., 27-30.

---

## Author Comment (AC2) · 11 Jun 2019

**Authors' response to referee comment amt-2018-460-RC2**
(**referee comments in bold font** and author's reponse in normal font):

**Review of the manuscript amt-2018-460 "True eddy accumulation trace gas flux measurements: proof-of-concept" by Lukas Siebicke and Anas Emad**

**The authors present a new setup for performing True Eddy Accumulation (TEA), where air is sampled separately for updrafts and downdrafts at a flow proportionally to the magnitude of vertical wind speed. This allows for measuring fluxes of constituents where no fast sensors are available that allow direct eddy covariance measurements. Other than Relaxed Eddy Accumulation (REA), TEA is a direct flux method and has therefore less theoretical limitations (e.g. scalar similarity, turbulent characteristics). The main technical advancement that allowed performing the TEA measurements was the development of a mass flow controller, that is capable of regulating the flow at 10 Hz to resolve the turbulent motion. Unfortunately, this work just gives examples of the performance of this new device, but no detailed technical descriptions or a more detailed description of the working principle of the mass flow controllers used. For the advancement of the TEA-technique, it is important that this information will be available in an appropriate way, soon.**
**Nevertheless, the authors are able to show the superior performance of their TEA-setup with respect to prior attempts in comparison to eddy covariance measurements and perform a thorough error analysis. Furthermore, they discuss in depth the major error sources and give an outlook on how to further improve the performance of the method. The manuscript is generally well written, well structured and the results are illustrated by a sufficient number of figures. Therefore, I support publication after considering the minor revisions given below.**

**General comments:**

**The manuscript was originally submitted by one author. Formulations need to be adapted (e.g. I => we).**

**AR:** Thank you for identifying that the change in the list of authors had not been reflected consistently in all cases in the text. We have changed this in the revised manuscript where appropriate.

**The authors showed that the flow controllers could reproduce the correct flow with respect to a reference thermal mass flow controller and that they can resolve 10 Hz, but what about a zero offset (leak)? I think it is important to show here that there is really zero airflow if the other channel is sampled.**

**AR:** We agree that the leak rate of the system must be zero or at least very small relative to the sample flow. The total system leak rate consists of the leak rate of its components, including the mass flow controller ("zero offset (leak)") as mentioned by the referee, but also pumps, tubing, fittings and air reservoirs.
From our leak testing, we know that the leak rate of the mass flow controller was very small relative to the total system leak rate. We had therefore not specifically addressed the MFC leak rate in the original version of the manuscript. We did however mention that leaks in other parts of the system (primarily leaks in air bags which developed over time but also leaks in fittings) were significant, and were one of the reasons for improved follow-up prototypes and experiments (see continuous-flow design).
To address the referees comment addressing specifically the MFC, we have now added information to the revised manuscript, quantifying the leak rate of the MFC, confirming that the combined leak rate of the part of the system operating under partial vaccum, i.e. including the mass flow controllers, filters, pumps, tubes and fittings, but excluding the air bags, expressed in terms of the leak rate relative to the average inlet flow rate, is very small, i.e. smaller than 0.008, which would result in less than 1% of corresponding flux uncertainty. The leak rate and flux uncertainty related to the MFC is again just a fraction of above estimate, considering the presence of other components contributing to the leak rate during the tests, e.g. pumps.

**Specific comments:**

**P3, L26: $R^2$ is the coefficient of determination**

**AR:** We have corrected the text accordingly.

**P8, point 5: As an idea for improvement:**

**Would it be possible to place a fast thermocouple into the sample airstream and correlate this to the sonic temperature? It might not work for real time correction of the sampling, but give an estimate on the magnitude of decorrelation for post processing.**

**AR:** You are pointing to an important issue of eddy accumulation (and EC to some degree) and we are actively seeking solutions and appreciate any suggestions towards correcting for the effects of decorrelation. However, currently there appears to be no obvious quantitative correction approach, considering the real-time sampling and the lack of co-spectral information.

We appreciate the suggestion of assessing decorrelation through the use of a thermocouple in the airstream. This, however, is complicated by the fact that the sonic temperature is not measured at a point in space and time but over a volume (height 11 cm, horizontal diameter 10 cm, path length 14.5 cm). More specifically, we know that wind and temperature measurements do not even represent an instantaneous volume average but result from several individual spatially discrete path averages, over discrete and lagged time periods (in total tens of milliseconds). At the same time, the spatial separation of the air inlet from the sonic thermometer measurement volume in our experiment was similar (18 cm vertical, 2 cm horizontal separation) to the sonic thermometer volume dimension itself (10, 11, 14.5 cm, see above). The horizontal inlet separation would thus be equivalent to a mere 14% of the sonic path length and the vertical separation equivalent to 124% of the path length.

Therefore we argue that, at the small spatial scales on the order of the sonic path lenth (i) it would be difficult to define a meaningful sonic temperature measurement to be used as reference for the suggested thermocouple measurement and (ii) the impact of mentioned decorrelation due to the small spatial separation of the air inlet is likely not exceeding or on the same order of magnitude as the impact of path averaging of the sonic itself or of typical open-path gas analyzers, and (iii) while decorrelation is related to flux errors it is not a sufficient measure to derive quantitative flux correction factors in the absence of high-frequency scalar measurements or cospectral information (see Horst and Lenschow, 2009).

Despite above noted lack of cospectral information and complications of the sonic sensing volume, we did nevertheless quantify temporal decorrelation based on our experimental data (Fig. 1). For this analysis we apply the simplifying assumptions that the wind measurements were obtained instantaneously and at the center of the sonic volume. We then calculated the travel times of the air over a distance corresponding to the separation of the air inlet from the center of the sonic anemometer, using 10 Hz wind measurements. In order to estimate the impact of the temporal decorrelation of the wind measurement and the air sampling on the scalar fluxes we weighted the air travel times by a factor proportional to the amplitude of the vertical wind velocity, analog to the procedure of scaling the physical sampling of individual 10-Hz samples which are subsequently mixed to form the average vertical velocity weighted 30-min air samples.

The resulting distribution of weighted decorrelation timings (Fig. 1) shows that the decorrelation times are generally larger for the vertical separation than for the horizontal separation. Regarding horizontal separation, the majority (84%) of decorrelation times are below the high frequency sampling period of 100 ms (10 Hz), which we assume to be sufficient to capture the majority of relevant turbulent scales. The decorrelation due to vertical separation is larger and frequently on the order of 500 ms. We expect that this would lead to some degree of flux attenuation. However it should be noted that the flux attenuation in horizontal and vertical directions is not symetric: data from the HATS experiment by Horst and Lenschow (2009) confirmed earlier findings „that flux attenuation is less with the scalar sensor located below the anemometer than if the scalar sensor is displaced an equal distance either horizontally or above the anemometer." It follows for the current experiment, where the air inlets were placed below the sonic sensing volume, that the vertial separation is relatively less important compared to the horizontal separation.

Figure 1: Density distribution of wind travel times over the spatial separation of the air inlet and the center of the sonic anemometer, separately for horizontal and vertical separation and corresponding horizontal (U) and vertical (w) wind components.

**P9, point 1: As the working principle of the mass flow controller has not been mentioned it is difficult to judge if there are issues in the mass flow control due to the effect of water vapor (e.g. (Lee, 2000). If there is a latent heat flux moisture in the updrafts and downdrafts must be different. Can the authors comment on how this could affect the volume mismatch?**

**AR:** Yes, a latent heat flux would lead to differences in the average water vapor mixing ratio of updrafts and downdrafts. Water vapor differences would also affect the sampling of the air via the effect of water vapor on air density (Lee 2000). However, as stated by Lee (2000), the proposed corrections only apply to the specific type of mass flow controller presented therein (thermal mass flow controller).
Regarding the current experimental setup, we have implemented a correction for the effects of water vapor on the mass flow sampling just after completion of the experimental period presented here. In order to quantify the uncertainty from water vapor differences regarding the current data set, we calculated the density of 30-min updraft and downdraft samples for the entire experiment and found that the difference between updraft and downdraft densities was up to 0.068%. This number is about two orders of magnitude smaller than the volume differences due to non zero 30-min mean vertical wind velocity (see Fig. 17 and Fig. 28), which relate to the volume mismatch correction (Eq. 10). Even if we assumed that the updraft samples were saturated with water vapor at much higher temperatures (40 degC) than present during the experiment, i.e. assuming a maximum typical value for the atmosphere of 4% water vapor (Lee, 2000) and at the same time assuming 0% moisture for the downdraft sample, which would result in an unrealistically large moisture difference, even then the air density of updraft and downdraft samples would only differ by 1.5% (density of 1.1105 kg m-3 at 4% vs. 1.1274 kg m-3 at 0%). The difference of 1.5% is still at least about one order of manitude smaller than the differences in mean vertical wind velocity leading to volume mismatch correction. Therefore we consider the impact of water vapor on the volume mismatch correction to be of minor importance relative to the vertical wind.

**P10, L5: Please provide coordinates.**

**AR:** The coordinates are provided in the revised version of the manuscript.

**P14, L16-18: Do not fully understand this sentence please consider revising.**

**AR:** We have rephrased the sentence as suggested and trust it is more clear now.

**P16, eq. 7:** If $\bar{w} = 0 \Rightarrow F_c = 0$. Therefore, it must be either $\left|\overline{w^+}\right|$ or $\left|\overline{w^-}\right|$ (as they are of equal magnitude. Please clarify.

**AR:** Please note that Eq. 7 of the original manuscript, which you reference, indicates to first take the absolute value of $w$ and then apply the temporal mean, which is denoted by the overbar, which includes the absolute value! This means the order of the mathematical operations is important! We believe that the manuscript is already correct. To avoid further misinterpretation, we have included a sentence to explicitly alert the reader.

**P17, eq. 10: Is** $\bar{w}$ **here the mean vertical velocity (for the averaging period) or is it the "mean of the absolute value of vertical wind velocity". Please clarify.**

**AR:** The first case is correct, i.e. $\bar{w}$ is „the mean vertical velocity (for the averaging period)". We have now explicitly stated this verbally in the revised manuscript in addition to the already correct mathematical notation.

**P18, first paragraph: Would be very valuable to have a schematic sketch of the setup explaining length of inlet lines, position of flow controllers and sampling bags including dead-volumes.**

**AR:** We have added a detailed technical description of the layout and functioning of the true eddy accumulation system as presented in a new Figure 5. By this we present the missing technical details als requested, addressing the referees questions about the length of inlet lines, position of flow controllers and sampling bags including dead-volumes. We go further by providing information about the detailed piping layout, position of all sampling system components, the operating pressures specifically for different sections of the system as well as the timing of the air transit through different parts of the system. The new information has been added in an appropriate place under section „2.8.3 TEA instrumentation and technical implementation".

**P20, L14: Which type of embedded computer?**

**AR:** We used an ARM-based single-board Linux computer of the type „Raspberry Pi" (Raspberry Pi Foundation, UK). The manuscript was updated accordingly.

**P25, L10: How was the reference signal generated? => Description was hidden in the figure caption of Fig. 8. Please include in text as well.**

**AR:** We have included the description in the text as suggested.

**P29: Please enlarge figures 13 and 14. In Fig. 13 the symbols, weather crosses or circles as described in the legend, are not identifiable. In Fig. 14 it is difficult to distinguish the different lines. Furthermore, from Fig. 14 a) it seems that TEA is systematically underestimating the fluxes during day, which is not so clear looking at panel b and especially the slopes of the linear regressions. Although I understand the reasoning for using standard major axis regression for two independent variables it would be interesting to see how a "standard" linear regression would look like. I think it can be justified to use the EC flux as the "controlled variable" by the statement that EC serves as a reference method.**

**AR:** Figures 13 and 14 have been enlarged, allowing to distinguish symbols in Fig. 13 and lines in Fig. 14.

Regarding the apparent differences between fluxes presented in subfigures a) and b) and your concerns of „systematically underestimating the fluxes during day" (a) and „the slopes of the linear regressions" (b) we would like to share the following observations:

1. The two subfigures are based on the same data set.
2. What the referee observed as an „understimation of fluxes during the day" in subfig a) is also present in subfig b) if examining the plot closely: while the linear model slope due to its averaging properties is close to one, the data points in b) show the same systematic behaviour seen in a), namely they follow on a „banana shaped" curve rather than a straight line, i.e. TEA fluxes tend to underestimate EC fluxes for medium flux amplitudes during the morning and the afternoon, while they are relatively high at noon and at night.
3. Please observe that there is a positive offset of 2.4 ymol m-2 s-1 rather than a slope anomaly, which is consistent for all TEA-EC regressions in subfig b). If you were to add this offset of 2.5 ymol m-2 s-1 to the TEA fluxes in a), denoted by the black line, then it would become more apparent that there is not a general slope issues but rather a non-linear relation ship between TEA and EC.
4. It is important to exercize caution in the interpretation of the TEA vs EC comparisons and in particular in interpreting the regression slopes due to the know deficiencies of the EC reference system (as explained in the manuscript). The slopes were found to be sensitive to the flux filtering by quality flags and EC-EC „consistency filter" (see text): the latter filter alone accounted for a slope change of 20% in the EC-EC regressions (see page 30, L32 with one EC sonic showing 18% lower fluxes without filtering and 2% higher fluxes with filtering, relative to the other EC sonic).
5. Given above mentioned uncertainties of the EC measurements, the current data set will not allow an ultimate comparison of TEA and EC flux measurements. To this end we have performed further experiments which confirmed that the issues observed in the current study to a large extend disappear when using a different and fully functional sonic anemometer and improved TEA flux sampler.
6. Regarding the question on how regression parameters of a „standard linear regression" would look like: we have fitted ordinary least squares (OLS), major axis (MA), and standard major axis (SMA) models. Results of the MA model are already reported in the manuscript (Fig. 14 b). Here we report the requested results for the OLS method (and for convenience also MA):

   TEA vs. EC („TEA sonic"):
   OLS: y = 0.91x + 1.81 with the 2.5% and 97.5% confidence intervals of the slope being 0.82 and 1.00, resp.

MA:  y = 1.01x + 2.44 with the 2.5% and 97.5% confidence intervals of the slope being 0.92 and 1.12, resp.

TEA vs. EC („EC sonic"):
OLS: y = 0.86x + 1.77 with the 2.5% and 97.5% confidence intervals of the slope being 0.77 and 0.95, resp.
MA:  y = 0.96x + 2.40 with the 2.5% and 97.5% confidence intervals of the slope being 0.86 and 1.06, resp.

TEA vs. EC (mean of „TEA sonic" and „EC sonic"):
OLS: y = 0.90x + 1.83 with the 2.5% and 97.5% confidence intervals of the slope being 0.81 and 0.98, resp.
MA:  y = 0.99x + 2.41 with the 2.5% and 97.5% confidence intervals of the slope being 0.90 and 1.09, resp.

It can be seen from the data that the OLS model slopes are for both sonic anemometers than the MA model slopes, as the referee might have expected from Fig. 14 a). However, we argue that OLS is not an appropriate model in this situation as it only considers errors in y and not in x. We cite Wehr and Saleska (2017): „The OLS fit line is unbiased only when there is negligible error in xˆ and when the error variance for the yˆ i does not vary with i. In this case, the problem reduces to minimizing the sum of the squares of the vertical distances of the points from the fit line". In the current case, there are non-negligable errors in x, therefore we chose MA, i.e. a type of „Model II" regression, which considers errors in both x and y. While we understand the and principally share the intention of the referee to use EC as the reference („I think it can be justified to use the EC flux as the "controlled variable" by the statement that EC serves as a reference method"), such declaratory statement does not solve the issue of error in the EC flux measurements, therefore we argue a model II type regression is more appropriate. In any case, both results are available now to the reader's convenience.

**P30, L19-21: From Fig. 14 it seems that this statement needs to be clarified. The two EC-flux estimates agree much better than the EC and the TEA measurements.**

**AR:** We appreciate the referees comment in the sense that apparently the statements indeed needed clarification. We have provided such clarification in the text of the revised manuscript. The important point, which we explicitly express in the revised text, is, that the EC-EC agreement is to some degree artificial as it results per definition from the application of what we refered to in the text as „consistency filter". What the filter does is it retains only those EC fluxes from the two setups which are closer to each other than a defined threshold. Therefore the two EC flux estimates can not  but match. The degree of matching becomes a function of the parameters choice of the similarity filter. Regardless, as stated in the text, method specific differences remain nevertheless but can not fully be separated from the filter effect given the current data set.

**P31, first paragraph: It would be good to name all cases always in the same order or mark them with indices.**

We double checked the order and believe the order in the text is consistent, namely:
1. TEA vs EC (TEA sonic)
2. TEA vs EC (EC sonic)
3. TEA vs. EC (average of TEA and EC sonic).
Note that „EC flux using TEA sonic" means EC flux not TEA flux. For improved clarity, we have introduced indices and brackets in the text to clearly convey this.

**P33, Fig. 15: In Panel b the legend is hardly distinguishable from the data-points.**

**AR:** The panels have been enlarged with more space, and the legend has been clearly separated.

**P34, last line: "akin"? Not clear what this word means in this context. Consider revising.**

**AR:** The text has been revised: previously „weights of updrafts and downdrafts in the mean akin to the asymmetry" has become „weights of updrafts and downdrafts in the mean which reflect the asymmetry". We hope the new version is more concise.

**P36, L6:"negative bias" => - 0.08**

**AR:** Thank you, we have corrected the sign.

**References:**
**Lee, X.: Water vapor density effect on measurements of trace gas mixing ratio and flux with a massflow controller, J. Geophys. Res. Atmos., doi:10.1029/2000JD900210, 2000.**

**Additional References in Author's response:**

Horst and Lenschow: Attenuation of Scalar Fluxes Measured with Spatially-displaced Sensors, *Boundary-Layer Meteorol (2009) 130:275–300*).

Richard Wehr and Scott R. Saleska, 'The long-solved problem of the best-fit straight line: application to isotopic mixing lines', Biogeosciences, 14, 17–29, 2017, doi:10.5194/bg-14-17-2017, www.biogeosciences.net/14/17/2017/

Sokal, R. R. and Rohlf, F. J.: Biometry: the principles and practice of statistics in biological research, 3rd Edn., W.H. Freeman and Co., New York, 1995.